# Multi-task Learning Long Short-term Memory Model to Emulate Wind Turbine Blade Dynamics

Shubham Baisthakur [1] and Breiffni Fitzgerald [1]

[1]School of Engineering, Trinity College Dublin, Ireland

**Correspondence:** Breiffni Fitzgerald (Breiffni.Fitzgerald@tcd.ie)

**Abstract.** The substantial computational expense associated with the dynamic analysis of wind turbines prohibits efficient design evaluations and site-specific performance predictions. This research explores the effectiveness of Principal Component Analysis and Discrete Cosine Transform dimensionality reduction methods to identify key spatial and temporal patterns in a wind field, which are subsequently used by a long-short-term memory (LSTM) algorithm to model the wind turbine responses. This study strikes a balance between prediction accuracy and training data requirements by employing an efficient feature selection technique and a multi-stage modelling approach which incrementally learns the information about the target variable. Furthermore, a multi-task learning strategy is adopted, allowing the LSTM model to predict multiple target variables at once, thus removing the necessity for separate models for each target variable. This method alleviates the computational cost of dynamic analysis of a wind turbine by addressing the challenges introduced by high-dimensional wind fields and time-consuming numerical integration processes. The findings show that this comprehensive approach significantly reduces computational cost while maintaining accuracy across all target variables, thereby facilitating design feasibility assessments and site-specific studies of wind turbines.

## 1 Introduction

The wind turbines, in an attempt to maximise energy capture, have grown significantly over the last few decades, with their scale seeing unprecedented growth (Roga et al., 2022). The increased scale of wind turbines translates to higher loads, deformations and more accumulated damage. Achieving an efficient design in the presence of these challenges is not a trivial task. Studies focused on efficient controls, advanced technologies, and an improved understanding of wind turbine operations have led to a more efficient operation of wind turbines (Sarkar et al., 2020b; Sarkar and Fitzgerald, 2020, 2022; Abbas et al., 2022; Njiri and Söffker, 2016; Sun et al., 2012; Tan et al., 2022; Fitzgerald et al., 2023). In addition, use of reliability and optimisation principles in the design of wind turbines can further improve the design ensuring consistent reliability in the wake of these challenges. However, evaluating multiple possible design combinations to arrive at an optimal solution satisfying multiple constraints requires high computational resources due to the non-linear dynamics of the underlying model and the non-convex nature of optimisation problem. Such optimisation studies can become computationally unfeasible to perform with an increasing number of parameters. For a wind turbine, the computational cost of optimisation is further compounded due to a large number of uncertain parameters involved, site-specific loading envelopes and the high computational cost of running numerical models.

To address these issues, this study presents a methodology to develop a machine-learning-based model to predict the dynamic response of a wind turbine at a fraction of the computational cost of a numerical model. The proposed model is not only efficient but also maintains high accuracy, as demonstrated through benchmarking against a validated numerical model, discussed later in this study.

Dynamic analysis of wind turbines refers to analysing the structural response subjected to stochastic wind inflow during operation. Wind speeds can fluctuate significantly across the rotor plane within the wind turbine rotors, resulting in varying wind conditions experienced at various points within the rotor area. This effect is more pronounced in turbines with higher rotor diameters. Furthermore, since wind speeds change over time, these spatial points within the rotor area are subjected to temporal variations in wind speed. Therefore, accurate prediction of a wind turbine's response to dynamic forces necessitates a realistic simulation of wind speed variations across its large rotor area and their evolution over time. The complexity of the spatio-temporal wind field introduces a fundamental challenge in wind turbine modelling in terms of a high dimensionality of the input space. TurbSim (Jonkman, 2009), a widely used turbulent wind field simulator, is a key tool for this purpose but is impacted by this dimensionality issue. The challenges arising from the high dimensionality of TurbSim data have been highlighted in many studies (Pereira et al., 2019; Haghi and Crawford, 2021; Bashirzadeh Tabrizi et al., 2019). Some studies have explored the use of surrogate modelling approach (Haghi and Crawford, 2023) and dimensionality reduction techniques (Lataniotis, 2019; Garcke et al., 2017) to alleviate this issue. In this study, the dimensionality reduction approach has been used to extract critical information from a high-dimensional representation of the wind field. Dimensionality reduction has been an active area of research in the domain of surrogate modelling (Hou and Behdinan, 2022), processing speech signals (Markaki and Stylianou, 2008), digital photographs (Van Der Maaten et al., 2009), or medical imagery (Hamarneh et al., 2011). Specialised literature on various dimensionality reduction techniques and their comparative performance on standard datasets is presented by Van Der Maaten et al. (2009). By identifying and retaining the most informative features, dimensionality reduction techniques facilitate efficient analysis of a dataset while reducing computational complexity and improving interpretability. To this end, Principal Component Analysis (PCA) and Discrete Cosine Transform (DCT) are used in this study to arrive at a low-dimensional representation of the inflow wind. Further, building on these extracted features, an LSTM model is developed to capture the temporal dependence between the features of inflow wind and structural response.

Long Short-term Memory (LSTM) Models are a type of recurrent neural network (RNN) with an internal memory state that captures the long-term dependence of the input features on the target variable. LSTM models have been implemented successfully in wind turbines for power forecasting (Banik et al., 2020; Yu et al., 2019; Woo et al., 2018) and damage detection (Choe et al., 2021; Xiang et al., 2021; Chen et al., 2021). Further, Dimitrov and Göçmen (2022) has demonstrated the use of LSTM as a virtual sensor, which can be used to predict the wind turbine parameters that are difficult to measure accurately on-site using SCADA and operational load measurements. Such models are applicable during the operational phase of wind turbines, where on-site measurements are available. However, very limited literature exists for the use of LSTMs in structural response prediction of wind turbines during the design and analysis stage, where extensive load measurement and SCADA data are not available (Woo et al., 2018; Shi et al., 2023; Zhu et al., 2024; Baisthakur and Fitzgerald, 2024). In this context, the

current manuscript aims to develop an LSTM model for application in the analysis and design stage, focusing on predicting the dynamic response of a wind turbine using the model-generated datasets.

Wind turbines represent a special class of structures whose response is impacted by multiple disciplines, including atmospheric modelling, principles of machines, structural dynamics, control engineering, and electronics. An efficient surrogate model should be able to integrate the various principles impacting the wind turbine while computing its response. To address this, a multi-stage modelling approach has been used where incremental information about the target is gained in multiple stages. Due to the increasing scale and flexibility of the wind turbines, multiple degrees of freedom are required to model a wind turbine structure and capture its intricate deformation patterns. Therefore, in order to get complete information about the system, the response at each DOF needs to be evaluated. However, creating a surrogate model for each DOF would necessitate developing multiple surrogate models. As the number of DOFs increases, the computational burden associated with training and implementing individual models can increase significantly. To address this, multi-output learning, also known as multi-task learning, has been used in this study to model multiple target variables using a single LSTM model (Thrun and Mitchell, 1995; Caruana, 1997). Multi-task learning leverages the inherent relationships between different target variables to develop a single, unified model capable of simultaneously predicting multiple target variables for a given set of input parameters. This approach is particularly useful in modelling structural response where multiple response variables are closely related to each other and are driven by a common external force. The use of dimensionality reduction techniques with a feature selection algorithm and multi-task learning approach leads to an efficient LSTM model capable of emulating the dynamics of wind turbines. The organization of the manuscript is as follows: Section 2 details the numerical model of the wind turbine employed in this research, while Section 3 offers a concise overview of the importance of dimensionality reduction and the methods implemented in this research. Section 4 elaborates on the LSTM architecture and the multi-task learning strategy. In addition, sections 5 and 6 describe the method for generating stochastic wind fields and the rationale for selecting input and output parameters for the surrogate model, respectively. Finally, Section 7 illustrates the numerical results, evaluating model performance regarding accuracy and computational efficiency. The manuscript concludes with Section 8, which summarises the key findings and contributions of this research.

## 2 Numerical Model of the IEA-15MW Wind Turbine

The numerical model of the wind turbine used in this study is developed using a multi-body dynamics methodology, based on Kane's dynamics principles (Kane and Levinson, 1985). Kane's dynamics approach is very effective in managing the intricate interactions among various components of the turbine, facilitating a precise representation of the overall system dynamics. By employing Kane's method, the complexity of deriving equations of motion is significantly reduced, and it allows for a simplified computer implementation compared to traditional methods such as Euler-Lagrange and D'Alembert's principle.

In total, 22 degrees of freedom are included to accurately represent the dynamics of the wind turbine components. The foundation is modelled as a monopile support structure with six degrees of freedom, encompassing three translational and three rotational motions. The tower is represented using the modal summation technique, which involves four principal mode shapes

that characterise the tower's movements in both the fore-aft and side-to-side directions. However, the axial shortening and twisting of the tower due to external loads are not taken into account. To ensure an accurate depiction of rotor speed, the azimuth of the generator and the twisting of the low-speed shaft are included in the model. The blades are treated as flexible elements, employing the modal summation method with three mode shapes for each blade—two modes for flapwise deformations and one mode for edgewise deformations. For force evaluation and modal integration, each blade is discretised into 50 stations along its span. This discretisation follows the original configuration provided by NREL in the release documentation of the IEA-15MW reference turbine (Gaertner et al., 2020).

Multiple reference frames are established to articulate the motion of different system components and to define their orientations relative to one another. The equilibrium equations for a simple holonomic multi-body system, derived using Kane's approach, are expressed as follows:

$$F_r + F_r^* = 0 \tag{1}$$

Here, $F_r$ represents the generalised active forces, while $F_r^*$ denotes the inertia force. These forces can be expressed in terms of kinematic variables as follows:

$$F_r = \sum_{i=1}^{n} {}^{E}v_r^{X_i} \cdot F^{X_i} + {}^{E}\omega_r^{N_i} \cdot M^{N_i} \tag{2}$$

$$F_r^* = -\sum_{i=1}^{n} {}^{E}v_r^{X_i}(m^{N_i} {}^{E}a^{X_i}) - {}^{E}\omega_r^{N_i} \cdot {}^{E}\dot{H}^{N_i} \tag{3}$$

In these equations, $F^{X_i}$ is the force vector acting on the center of mass of point $X_i$, and $M^{N_i}$ is the moment vector acting on the rigid body $N_i$. The terms ${}^{E}v_r^{X_i}$ and ${}^{E}\omega_r^{N_i}$ indicate the partial linear and angular velocities of point $X_i$ and rigid body $N_i$, respectively. Additionally, ${}^{E}\dot{H}^{N_i}$ represents the time derivative of the angular momentum of rigid body $N_i$ about its center of mass $X_i$ in the inertial frame, expressed by the equation:

$$ {}^{E}\dot{H}^{N_i} = \bar{\bar{I}}^{N_i} \cdot {}^{E}\alpha^{N_i} + {}^{E}\omega^{N_i} \times \bar{\bar{I}}^{N_i} \cdot {}^{E}\omega^{N_i} \tag{4}$$

The final governing equation of the system is structured as follows:

$$M(q,t)\ddot{q} + f(q,\dot{q},t) = 0 \tag{5}$$

In this expression, $M(q,t)$ denotes the inertia matrix, while $\ddot{q}$ is the acceleration vector. The function $f(q,\dot{q},t)$ represents the force vector, which comprises both external and restoring forces acting on the structure. Numerical methods are employed to solve this system of equations, specifically the fourth-order Runge-Kutta method in this study. Since the main objective of this numerical model is to generate the data required for developing the LSTM model, a comprehensive derivation of equations of motion is not presented in this work, but interested readers may refer to Sarkar and Fitzgerald (2021) for further details. The application of this numerical model across multiple domains has already been established in prior studies (Sarkar et al., 2020b, a; Sarkar and Fitzgerald, 2020, 2022; Fitzgerald et al., 2023)

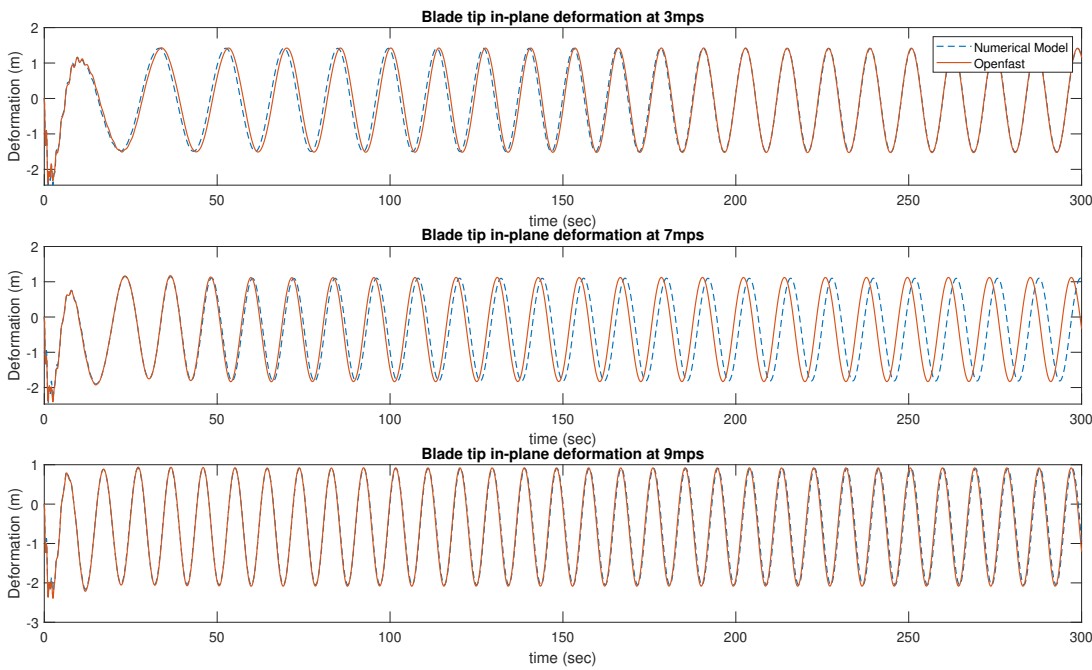

**Figure 1.** Model Verification: In-plane deformation response at the blade tip

The analysis conducted in this research employs the specifications of the 15MW wind turbine from the International Energy Agency (IEA), categorised as an International Electrotechnical Commission (IEC) Class 1B direct-drive unit, with a rotor diameter of 240 m and a hub height of 150 m. The technical report outlining the IEA 15-megawatt reference wind turbine Gaertner et al. (2020), provides a comprehensive explanation of the wind turbine's key characteristics and operational parameters.

The wind turbine is modelled on the basis of Kane's dynamics principles. The multibody model developed in this research has been validated by comparing its response to the OpenFast model of the National Renewable Energy Laboratory (NREL) (Jonkman et al., 2024), a widely used open-source framework for wind turbine dynamics simulation. The validation is performed for the below-rated steady-state conditions. The steady-state conditions are used as a baseline for validation, as it simplifies the analysis by isolating the model's behaviour from the added complexity introduced by transient dynamics. This validation exercise verifies the overall stiffness of the model and ensures that it accurately captures the fundamental structural and aerodynamic interactions. The results of this validation are shown in Fig. 1, 2, 3 and 4. Fig. 1 and 2 present a comparison of the blade tip deformation response between the developed numerical model and OpenFAST, in the in-plane and out-of-plane directions, respectively. Fig. 3 shows the comparison of the rotor speed, while Fig. 4 illustrates the comparison of the FFT of the side-to-side deformation of the tower top between the two models. The comparison highlights that the numerical model developed in this study can accurately match the response prediction of the OpenFast model, establishing the accuracy of the numerical model. This model is utilised to generate the data required for training the machine learning model.

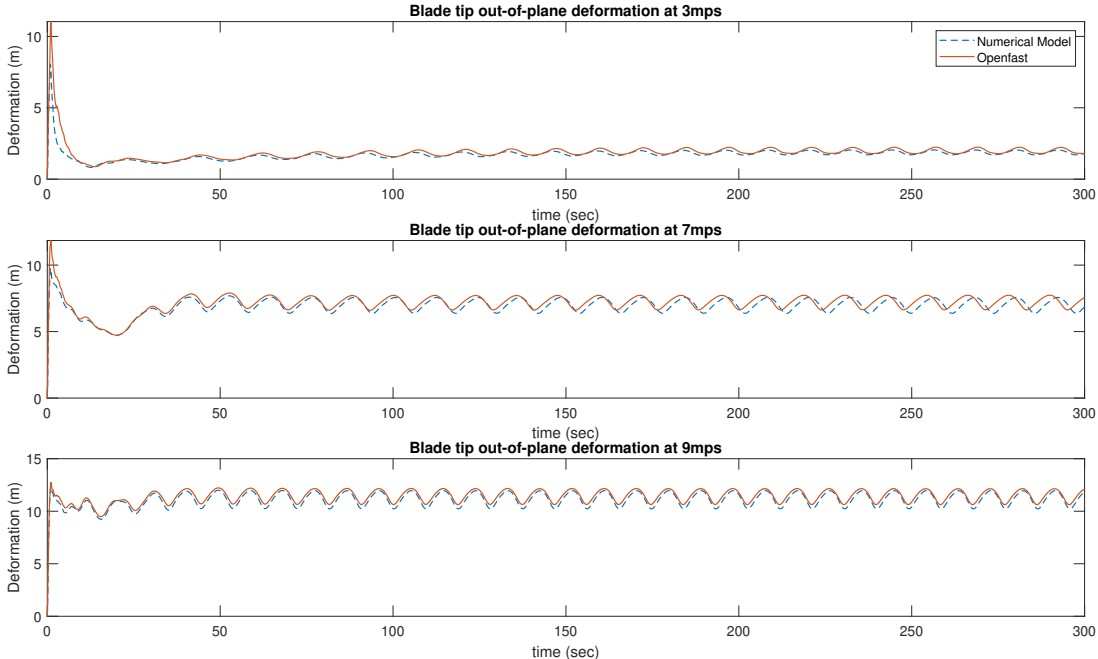

**Figure 2.** Model Verification: Out-of-plane deformation response at the blade tip

This numerical model uses the turbulent wind inflow generated by TurbSim to drive the system response. Wind fields created by TurbSim offer a high-dimensional representation of turbulent wind. The following section presents the dimensionality reduction techniques employed to tackle the high dimensionality of the wind-field data.

## 3    Dimensionality Reduction Techniques for Wind Field Data

Dimensionality reduction refers to the process of transforming high-dimensional data into a meaningful representation with reduced dimensionality. Dimensionality reduction is often used as a preprocessing step before building surrogate models for high-dimensional input spaces. Reduction in dimensions facilitates faster model training and can improve the robustness of surrogate models by focussing on the most relevant features (Bishop and Nasrabadi, 2006; Brunton and Kutz, 2022).

Mathematically, let $\mathbf{X}$ denote a high-dimensional dataset, where each row corresponds to an observation and each column represents a feature. If $p$ is the number of observations and $N$ is the number of features, then $\mathbf{X}$ is an $p \times N$ matrix. If $p$ is a large number, handling such high-dimensional data poses computational challenges and can lead to inefficiencies in analysis and modelling tasks. Let $\mathbf{Z}$ represent the reduced-dimensional representation of the original dataset $\mathbf{X}$, where $\mathbf{Z}$ is an $p \times n$ matrix with, where $n << N$ being the reduced number of dimensions. The goal of reducing dimensionality is to find a mapping function $f : \mathbf{X} \rightarrow \mathbf{Z}$ that captures the important information in the original data while reducing its dimensionality.

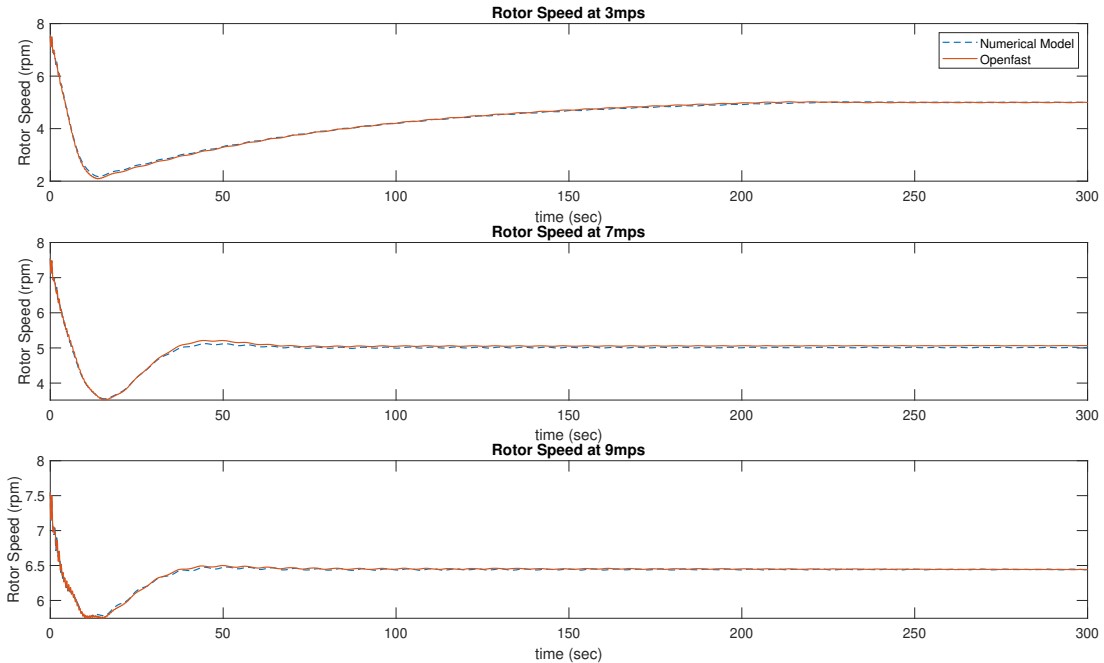

**Figure 3.** Model Verification: Comparison of rotor speed

In the context of wind turbines, TurbSim simulations can generate wind fields consisting of thousands of data points across space and time. Dimensionality reduction is crucial to transform these complex wind fields into more manageable representa-
tions for efficient response prediction. Various dimensionality reduction techniques are explored in the literature for application to high-dimensional problems; among all these techniques, Principal Component Analysis (PCA) is one of the most widely used approaches (Pearson, 1901; Jolliffe and Cadima, 2016). PCA is a statistical technique that focusses on capturing spatial correlations within the data by identifying the principal components that capture the maximum variance in the dataset. PCA offers computational efficiency and linear mapping, making it suitable for handling large-scale wind data sets encountered in
wind turbine modelling. Lataniotis (2019) has shown that PCA consistently outperformed the other dimensionality reduction techniques implemented in their research in terms of reconstruction error and robustness of results over different repetitions on the standard datasets used in dimensionality reduction problems. PCA implemented for wind speed forecasting (Skittides and Früh, 2014; Geng et al., 2020), wind turbine fault detection (Zhang et al., 2021), and monitoring (Wang et al., 2016) has also delivered good results. Principal components can capture recurring spatial and temporal trends in wind speed variations.
The reduced-dimensional representation using PCA further helps in data handling and analysis. While PCA excels at captur-ing the variance within the data through linear relationships, it might overlook the presence of underlying non-linear patterns in complex wind fields. Also, while principal components represent significant variance, interpreting their physical meaning

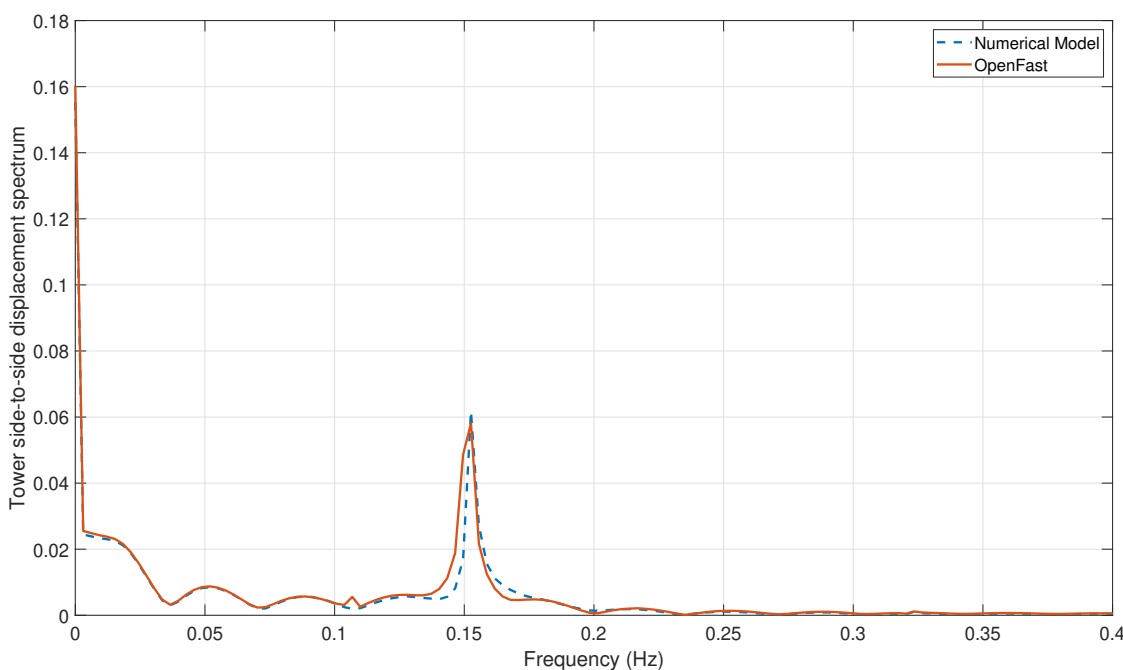

**Figure 4.** Model Verification: Comparison of Fourier spectrum of tower response

can be challenging for wind fields. To address these potential limitations, the Discrete Cosine Transform (DCT) is used as a complementary technique in this research.

The Discrete Cosine Transform (DCT) offers a powerful tool to analyse the dynamics of the wind field by decomposing the data into its underlying frequency components, revealing patterns of spatial and temporal variation (Ahmed et al., 2006). Unlike PCA, DCT emphasises the frequency-domain representation of signals or images, using its energy compaction property to highlight dominant frequency components. This data transformation allows DCT to capture the energy content of a signal or image in fewer coefficients by exploiting the characteristics of the data in the frequency domain. This property enables effective

compression and representation of wind data, which can be exploited for dimensionality reduction by capturing dominant spatial and temporal correlations inherent in wind speed fields. The components obtained through DCT (cosine functions with specific frequencies) correspond directly to spatial variations of different wavelengths or scales within the wind-field data. This representation makes the DCT components physically interpretable. Recently, DCT has been applied by Schär et al. (2024) for dimensionality reduction of the stochastic wind field and has been shown to deliver good results.

Given the distinct strengths and limitations of both PCA and DCT for wind field representation, their suitability to reduce the dimensionality of turbulent wind fields and capture maximum information with a minimum number of variables is investigated in this study. The mathematical formulation of these methods is presented in the next section.

### 3.1 Principal Component Analysis

PCA is a mathematical tool that transforms potentially correlated features into a smaller set of uncorrelated variables called principal components. This transformation maximises the variance explained by each component, thus highlighting the most prominent patterns within the data. In PCA, the data is normalised to have zero mean to ensure that principal components capture variations from the average behaviour and not the absolute magnitude of the wind speeds. Following the original notations, assume ($\mathbf{X} \in \mathbb{R}^{p \times N}$) represent the original wind field dataset matrix, with $p$ observations (time steps) and $N$ features (spatial locations). The mean-centred dataset ($\mathbf{X}' \in \mathbb{R}^{p \times N}$) is given by:

$$\mathbf{X}' = \mathbf{X} - \overline{\mathbf{x}} \tag{6}$$

where $\overline{\mathbf{x}}$ is the mean vector of the dataset. PCA computes the covariance matrix to quantify the pairwise linear relationships between different features within the wind field data. The covariance matrix ($\mathbf{\Sigma}$) of the mean centred data is computed as:

$$\mathbf{\Sigma} = \frac{1}{\mathbf{p}-1}(\mathbf{X'^{T} X'}) \tag{7}$$

Further, eigen-decomposition of the covariance matrix ($\mathbf{\Sigma}$) is performed to identify the directions of maximum variance within the wind field data and quantify the amount of variation captured along each direction. The eigenvectors ($\mathbf{V} \in \mathbb{R}^{N \times N}$ and their corresponding eigenvalues ($\mathbf{\Lambda} \in \mathbb{R}^{1 \times N}$ are represented using the eigenvector matrix $\mathbf{V}$ with $N$ columns and eigenvalue vector $\mathbf{\Lambda}$ with $N$ entries:

$$\mathbf{V} = \begin{pmatrix} \mathbf{v}_1 & \mathbf{v}_2 & \cdots & \mathbf{v}_N \end{pmatrix}$$

$$\mathbf{\Lambda} = \text{diag}(\lambda_1, \lambda_2, \ldots, \lambda_N)$$

In PCA, the eigenvectors are sorted in decreasing order of their corresponding eigenvalues so that the eigenvector with the largest eigenvalue represents the principal component explaining the most variance in the wind field data. The number of principal components required to capture the maximum information about the wind field data is examined using the cumulative explained variance calculated as:

$$e_k = \frac{\sum_{i=1}^{k} \lambda_i}{\sum_{i=1}^{N} \lambda_i} \tag{8}$$

A common approach is to select the smallest number of principal components ($k$) that achieve a desired percentage of explained variance. Finally, the original high-dimensional dataset $\mathbf{X}$ is projected onto the new basis spanned by the selected principal components to obtain the reduced-dimensional representation $\mathbf{Z_{PCA}} \in \mathbb{R}^{p \times n}$, where $n$ is the desired number of principal components to retain. The projection can be expressed as:

$$\mathbf{Z_{PCA}} = \mathbf{X}' \mathbf{V}_n$$

where $\mathbf{V}_n$ is the matrix containing the first $n$ eigenvectors corresponding to the largest $n$ eigenvalues. This projection transforms the original features representing wind speeds at different grid points into a reduced set of features. By selecting the $n$ principal components with the largest eigenvalues, PCA ensures that these new features capture most of the essential variation within the wind field data.

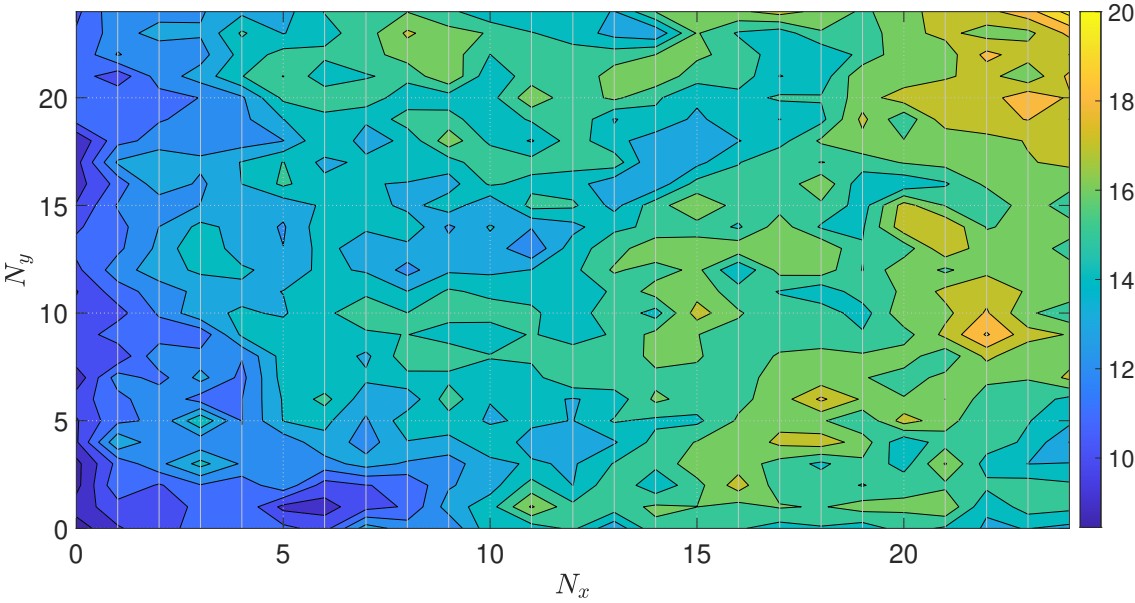

**Figure 5.** Modelling wind speed data as an image at a time instance t

## 3.2 Discrete Cosine Transform

The Discrete Cosine Transform (DCT) is a frequency-domain transformation technique which uses real-valued cosine functions as its basis. For a discrete signal $x(n)$ of length N, the 1D DCT is calculated using the Eq. 9

$$z(k) = \sum_{n=0}^{N-1} x(n) \cdot \cos\left(\frac{\pi}{N}\left(n+\frac{1}{2}\right)k\right) \tag{9}$$

where $z(k)$ is the $k$-th frequency component of the signal transformed using DCT. Following the approach implemented by (Schär et al., 2024), a mathematical correspondence can be established between a 2D spatial wind field grid generated using TurbSim and an image. At a given time instance $t$, the wind field is defined over a spatial grid of size $(N_x \times N_y)$, each grid point $(i,j)$, representing a spatial location in the wind field, can be mapped to a pixel in the image. The magnitude of the wind speed at a point in the grid, denoted by $(X(i,j))$, determines the intensity of its corresponding pixel (see Fig. 5). This formulation allows us to treat the spatial wind field at each time step as a 2D image, enabling the DCT to be applied independently at every time instance. This results in a time-resolved representation of spatial wind structures across the wind field. Using this representation of the wind field, 2D DCT is used to analyse wind speed variations across spatial scales and directions. The 2D DCT of the wind field $(X)$, denoted by $Z_{DCT}(u,v)$, is calculated using the Eq. 10:

$$Z_{DCT}(u,v) = \alpha(u)\alpha(v) \sum_{i=0}^{N_z-1} \sum_{j=0}^{N_y-1} X(i,j) \cos\left(\frac{\pi(2i+1)u}{2N_z}\right) \cos\left(\frac{\pi(2j+1)v}{2N_y}\right) \tag{10}$$

such that,

$$
\alpha(u) = \begin{cases} \frac{1}{\sqrt{N_z}}, & i = 0 \\[2ex] \sqrt{\frac{2}{N_x}}, & 1 \le i \le N_z - 1 \end{cases}
\tag{11}
$$

and

$$
\alpha(v) = \begin{cases} \frac{1}{\sqrt{N_y}}, & j = 0 \\[2ex] \sqrt{\frac{2}{N_y}}, & 1 \le j \le N_y - 1 \end{cases}
\tag{12}
$$

where $u$ and $v$ are frequency indices representing spatial frequencies in the transformed domain and $\alpha(u)$ and $\alpha(v)$ are normalisation factors ensuring orthogonality of the basis functions. The coefficients $(Z_{DCT}(u,v))$ obtained from the 2D DCT transformation represent the strength of different frequency components present within the wind field. These frequency components correspond to variations in wind speed at different spatial scales. The low frequency coefficients, associated with low values of $u$ and $v$, represent spatial patterns or trends throughout the wind field grid, which capture smooth variations in the spatial domain. In contrast, high-frequency coefficients represent finer-scale, localised wind speed fluctuations and turbulent structures. These coefficients capture rapid, often less spatially organised changes in wind speed over short distances within the wind field. The DCT transform must be applied at each time step, where the snapshot of the wind speed grid at that time instance acts as an image. By analysing the energy distribution across different DCT coefficients, the dominant spatial scales of variation present in the wind field can be identified. Using the 2D DCT of a wind field and ranking the coefficients by their magnitudes, a subset of coefficients that capture the majority of the essential variations of the wind field can be identified. Retaining the significant coefficients and discarding the high-frequency components results in a compressed representation of the wind field, preserving the dominant spatial patterns in the wind field. For each selected DCT component, its temporal evolution, formed by concatenating its values across successive time steps, constitutes a feature sequence. Further, an LSTM model is developed using these features to predict the deformation response of the wind turbine blades. Using these features for model training ensures that both spatial and temporal dynamics are effectively represented. This approach leverages the spatial structure of the wind field while preserving its temporal evolution. The details of the LSTM model architecture are presented in the next section.

## 4 Long Short-term Memory Model and Multi-task learning

An LSTM model is a type of Recurrent Neural Network (RNN) specifically designed to handle sequential data by capturing both long-term and short-term dependencies between input and output variables. Unlike conventional feedforward neural networks, which process input independently and lack memory of previous states, RNNs employ a feedback mechanism in which the network's output at each step is fed back as input. This recursive structure enables RNNs to model the temporal

dependencies inherent in sequential data. However, traditional RNNs suffer from the problem of vanishing gradients, where gradients become too small to effectively update the parameters of the network during training, particularly for long sequences (Hu et al., 2018). The LSTM architecture addresses this limitation by introducing gating mechanisms (input, forget, and output gates) that regulate the flow of information and gradients through the network (Hochreiter and Schmidhuber, 1997). These gates allow LSTMs to retain important information over long sequences and mitigate the vanishing gradient problem, making them well-suited for tasks involving sequential data with complex temporal dependencies. Although more advanced architectures such as Transformers (Vaswani et al., 2017) have gained popularity for their ability to model long-range dependencies through self-attention mechanisms and their parallelisable structure, they often require significantly larger datasets and computational resources to achieve optimal performance. Transformers excel in tasks like natural language processing, where massive datasets are available, and parallel computing can be fully leveraged. However, for applications such as wind turbine modelling, where data sets may be limited and the temporal structure of the data is critical, LSTMs offer a practical and efficient alternative (Li et al., 2018). Their ability to capture temporal dependencies with fewer parameters and computational requirements makes them a suitable choice for this specific domain. The core component of an LSTM is the memory cell, which consists of these gating mechanisms. Selective data retention and elimination are performed as the data passes through these gates. Based on the task of each gate, they are commonly known as

- Input gate: This gate receives the model features at the current time step and the hidden state predicted at the previous time step as input. This gate combines these two inputs to create a candidate for storing new information. Mathematically, this operation is represented in Eq. 13

$$i_t = \sigma_g(W_i \times x_t + R_i \times h_{t-1} + b_i) \tag{13}$$

- Forget gate: The input to the forget gate is the same as the input gate. This gate combines the information from input features at the current time step with the hidden state from the previous time step and estimates what past information is no longer relevant. This operation is presented in Eq.14

$$f_t = \sigma_g(W_f \times x_t + R_f \times h_{t-1} + b_f) \tag{14}$$

- Cell state: The cell state in an LSTM unit undergoes updates influenced by the output of the forget gate and the input gate, which determine the significance of new and existing memory. A cell state is the actual memory of the LSTM, which holds the information across many time steps. The cell state acts as the long-term memory of the unit. This operation is represented through Eq. 15 and Eq. 16

$$c'_t = \sigma_g(W_c \times x_t + R_c \times h_{t-1} + b_c) \tag{15}$$

$$c_t = f_t \cdot c_{t-1} + i_t \cdot c'_t \tag{16}$$

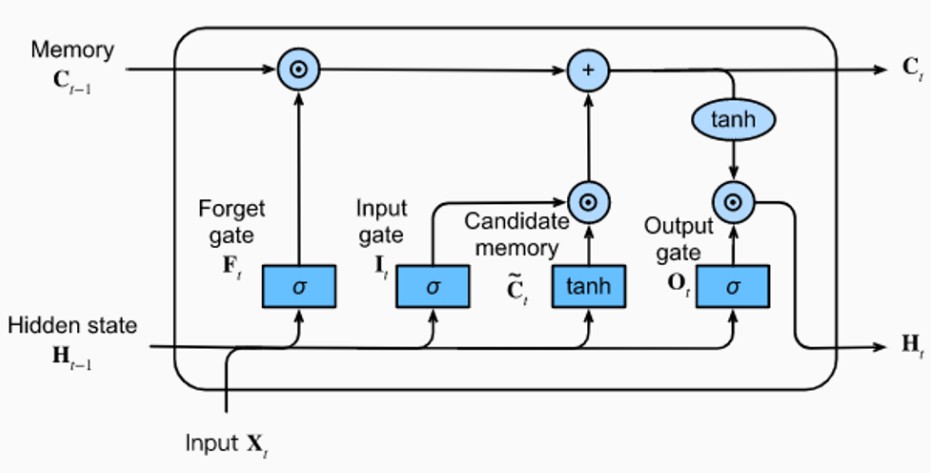

**Figure 6.** LSTM Architecture (adopted from Ref. (Calazone, 2022))

- Output gate: The output gate combines the cell state updated at the current time step with the previous hidden state, creating the current hidden state, predicting the network's output (hidden state) at the current time step. This gate acts as the short-term memory of the unit. Eq. 17 and Eq. 18 represents this operation mathematically

$$o_t = \sigma_g(W_o \times x_t + R_o \times h_{t-1} + b_o) \tag{17}$$

$$h_t = o_t \cdot \sigma_c(c_t) \tag{18}$$

In Eqs. 13 through 18, $x$ represents the feature vector of the LSTM model, while $h$ denotes the hidden state, a crucial output of the LSTM unit. The subscript $t$ denotes the temporal position of the vectors, where $t$ refers to the current time step, $t-1$ refers to the previous step, and so forth. The symbols $i$, $f$, $c$, and $o$ correspond to the input gate, forget gate, cell state, and output gate, respectively. In each gate, $W$ signifies the fixed weights, $R$ denotes the recurrent weights, and $b$ represents the bias term. The $\sigma$ symbol signifies the sigmoid activation function, ensuring non-linearity in the input-output transformation and constraining the gate's output within the interval [0,1]. The sigmoid activation function is given by Eq. 19

$$\sigma(x) = \frac{1}{1+e^{-x}} \tag{19}$$

Multiple LSTM units can be stacked together to capture complex relationships in the data. A sketch of a typical LSTM memory cell and data flow is presented in the Fig 6, adopted from Ref. (Calazone, 2022).

Multi-task learning is an approach in machine learning in which a model leverages the information shared between different output variables to learn the dependencies to predict multiple outputs using a set of common input parameters (Caruana, 1997). The multi-task learning model has been found to deliver better by jointly learning various dependent parameters than by learning them independently (Zhang and Yang, 2018). In the multi-task learning method, various deep learning layers are

stacked together to efficiently learn patterns from the data. In standard machine learning approaches, only one output parameter is learnt at once; however, multiple parameters equalling the total number of neurons in the final layer can be predicted using a multi-task learning model. An overview of multi-task learning, its application, and detailed classification can be found in (Zhang and Yang, 2018).

## 5 Generation of Stochastic Wind Field and Data Preparation

Wind turbines are subjected to stochastic turbulent wind inflow during their operation, which significantly influences power generation and structural loads. Generating a realistic wind inflow pattern is therefore crucial for ensuring the validity of simulation results. In this study, TurbSim (Jonkman, 2009) is used to generate the wind field that acts on the wind turbine. TurbSim serves as a stochastic tool to generate full-field representations of turbulent winds, using statistical models to emulate wind dynamics. TurbSim operates on a statistical model to produce time series data comprising three-dimensional wind speed vectors across a fixed two-dimensional vertical grid. This grid remains stationary in space during the inflow generation process. To generate wind fields specific to wind turbine configurations and site conditions, TurbSim provides customisable parameters that encompass turbine geometry and meteorological factors. These parameters control the size and complexity of the wind dataset. Key TurbSim parameters include:

1. **Random Seeds**: TurbSim uses two random seeds (*RandSeed1* and *RandSeed2*) to create random phases for the velocity time series and ensure reproducibility. These seeds generate unique stochastic realisations for specified environmental conditions. Further, these seeds can be varied at constant meteorological parameters to produce various stochastic patterns with similar statistical properties.

2. **Grid Dimensions**: The parameters *GridHeight* and *GridWidth* define the vertical and horizontal extent of the computational domain, while *NumGridY* and *NumGridZ* govern the spatial resolution. The hub height (*HubHt*) serves as a reference point for the placement of the grid. A representative image of the TurbSim grid encompassing the wind turbine rotor is presented in Fig. 7

3. **Meteorological Conditions**: The variable *TurbModel* defines the spectral model used for generating the wind speed field. Key statistical properties include mean wind speed (*Uref*), turbulence intensity (*IECturbc*), surface roughness length ($Z_0$), and power law exponent (*PLExp*).

The IEA-15MW reference wind turbine, characterised by a 240m rotor diameter and a 150m hub height, is modelled using a rectangular domain measuring 285m x 285m with a 25 x 25 spatial grid layout. A single 10-minute wind field, with a temporal discretisation of 0.05 seconds, yields 625 features with 12,000 observations each, resulting in a very high-dimensional input space. The dimensionality of the input space is further compounded, as multiple wind fields are required to train a surrogate model. An LSTM model may encounter challenges, such as overfitting and increased computational and memory requirements when dealing with such high-dimensional input data. To mitigate these issues, dimensionality reduction techniques are used to extract essential information from the input space while minimising the number of variables.

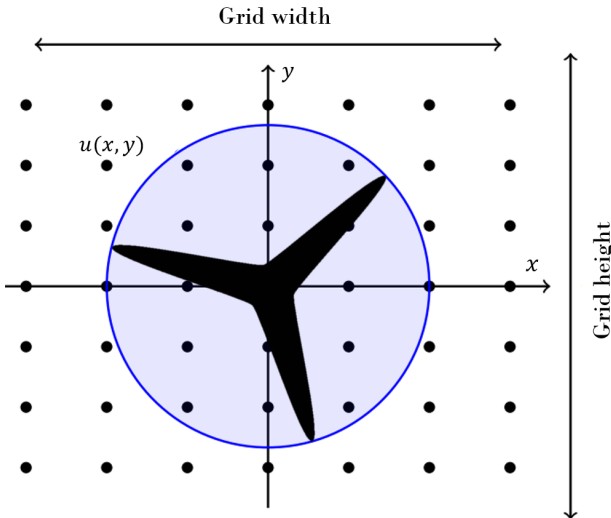

**Figure 7.** Visual representation of TurbSim Grid

## 5.1 Selection of Input Variables

The development of a surrogate model relies on the generation of a representative dataset covering various operational scenarios. Since the model learns the relationship between input and output purely based on the training data, the selection of wind field parameters and their distributions is critical. Although the surrogate model can extrapolate beyond the training data bounds, its prediction accuracy in these regions is uncertain (Chen et al., 2018; Hastie et al., 2009; Brunton and Kutz, 2022). This study focusses on onshore wind turbines operating under normal conditions, which significantly influence fatigue loading and power production. The inflow wind patterns are influenced by multiple parameters; however, modelling all of them is computationally prohibitive. Sensitivity analyses, such as those conducted by Dimitrov et al. (2018), have shown that turbulence intensity ($TI$), mean wind speed ($U$), and power law exponent ($\alpha$) have the greatest impact on wind turbine loads under normal operating conditions. These parameters are therefore selected for generating the wind fields. The bounds for these parameters are presented in Table 1. The wind speed range of 3–25 m/s is adopted from the operational range of the reference 15 MW wind turbine, ensuring that all simulations remain within its cut-in and cut-out wind speed limits. The bounds for $TI$ and $\alpha$ are based on the approach outlined in Ref. (Dimitrov et al., 2018). In this study, the lower limit for $\alpha$ is set at zero to focus the analysis solely on positive wind shear. Negative shear values were omitted to simplify the analysis and constrain the parameter space. Negative shear leads to uncommon inflow conditions that can cause atypical blade loading patterns that, although intriguing, fall outside the current surrogate modelling framework aimed at typical and frequently encountered operating scenarios.

The spatial correlations, temporal variability, and turbulence inherent in wind make it a fundamentally random process. This randomness stems from aleatory uncertainty due to the inherent variability of wind speed and epistemic uncertainty arising

| Parameter | Lower bound | Upper bound |
|---|---|---|
| Mean wind speed (m/s) | 3 | 25 |
| Turbulence intensity (%) | 0.025 | $\frac{0.18}{U}\left(6.8 + 0.75 \cdot U + 3\left(\frac{10}{U}\right)^2\right)$ |
| Shear exponent | 0 | $\alpha_{ref,UB} + 0.4\left(\frac{R}{z}\right)\left(\frac{U_{max}}{U}\right)$ |

**Table 1.** Environmental parameters used and their limits

from limitations to model or measure all factors influencing wind behaviour. Training a surrogate model capable of accurately predicting wind turbine responses under these uncertainties requires the generation of multiple realisations (ensembles) for each set of environmental parameters. Quantifying these uncertainties is essential for making realistic risk assessments. However, identifying the minimum number of seeds required so that the surrogate model can efficiently identify the underlying patterns in these wind fields is a crucial factor governing the amount of data required for training these models. In a different application, the IEC guidelines (IEC, 2019) recommend considering six random realisations for each mean wind speed to calculate fatigue damage, which implies that six seeds can effectively capture the variability in wind fields. In addition, these guidelines recommend keeping environmental conditions constant within a wind speed limit. This approach, combined with the inherent variability in $TI$ and $\alpha$, leads to a large number of required simulations. A study by Hübler et al. (2018) shows that assuming constant environmental conditions within a wind speed bin does not fully capture the uncertainty in fatigue damage and recommends scattering environmental conditions within a bin for a more comprehensive assessment. In this study, a combination of these sampling approaches is used to simulate the wind fields required to train the surrogate model; the mean wind speed values within the turbine's operational range are selected using Sobol sampling, a low-discrepancy quasi-random method that ensures uniform coverage of the wind speed space. Scattering environmental conditions within a wind speed bin also acts as a crucial data augmentation technique for the surrogate model, which prevents overfitting to a limited set of wind patterns and significantly improves generalisation. To this end, in this study a total of 16 realisations are generated for each mean wind speed $U$ in this study, assuming a random combination of *RandSeed1*, *TI*, and $\alpha$. To this end, in this study, a total of 16 realisations are generated for each mean wind speed $U$, assuming a random combination of *RandSeed1*, *TI*, and $\alpha$. In total, this approach results in 50 mean wind speed values each combined with 16 realisations (varying *RandSeed1*, *TI*, and $\alpha$), generating a dataset of 800 simulations for training the surrogate model. This ensures a broader sampling of wind patterns while considering computational constraints. As such, wind fields are generated using TurbSim with the following key settings:

1. Spectral Model: Kaimal spectral model (IECKAI)

2. Turbulence Model: Normal Turbulence Model (NTM)

3. Grid Size: 25 x 25 grid with a spatial extent of 285m x 285m

4. Temporal discretisation: 0.05 seconds

The response of the wind turbine for these loading conditions is simulated using the numerical model presented in Section 2 with a time step of 0.005 seconds to ensure accuracy and numerical stability. The response is simulated for 800 seconds, and the data for the first 200 seconds is excluded from the analysis to eliminate the effect of transience. The ROSCO controller (Abbas et al., 2021, 2022) is used to optimise the performance of the wind turbine by regulating the rotor speed and blade pitch angle. The structural response is then downsampled to align with the temporal resolution of the wind field data. The dataset is split into training, validation, and testing subsets with a ratio of 75:12.5:12.5. To ensure representative coverage across all mean wind speeds, this division is performed using the random seed index: for each mean wind speed, 12 wind fields are used for training, while 2 each are reserved for validation and testing. This approach preserves the statistical diversity across the datasets and prevents bias toward any particular wind condition.

The high dimensionality of the wind field data (a 25 x 25 grid over 600 seconds results in an input matrix of 12000×625) underscores the importance of dimensionality reduction techniques. The PCA and DCT methods transform a wind field grid into different representations, providing a different basis for capturing the spatio-temporal variations. However, these techniques alone do not reduce the number of features. To address this, the Recursive Feature Addition (RFA) method (Guyon and Elisseeff, 2003; Guyon et al., 2002) is used to select an optimal subset of features. RFA iteratively adds features based on their relevance to the target variable, evaluated using metrics such as root mean squared error (RMSE). This process continues until adding extra features no longer improves model performance. A detailed analysis of various approaches for feature selection, their computational cost, and resulting accuracy for an LSTM model is presented in Ref. (Baisthakur and Fitzgerald, 2025). By combining PCA, DCT, and RFA, this study identifies the most influential parameters for training the LSTM model, ensuring computational efficiency and robust performance.

## 6 Selecting the output parameter

The versatility of a surrogate model and its ability to provide comprehensive information depend on its output parameters. Within the existing literature, a variety of surrogate models have been formulated to predict quantities associated with load components, such as fatigue damage equivalent load and reactions in tower and blade structures (Haghi and Crawford, 2023; Schär et al., 2024; de N Santos et al., 2023; Bai et al., 2023). However, these load components are derived from responses at particular degrees of freedom (DOFs), and the extrapolation from DOF responses to load reactions remains straightforward, thus not necessitating substantial computational resources. In this specific context, the model developed in this study is aimed at predicting the dynamic responses of blade deformation and individual DOF which can be subsequently utilised to compute the loads and reactions. Moreover, to obtain results that are physically quantifiable, an additional LSTM model has been developed to predict the blade's in-plane and out-of-plane deformation responses. This selection of target variables ensures that the surrogate model maintains sufficient versatility to calculate multiple derived quantities.

In this approach, multi-task learning streamlines the modelling process and enhances the model's capacity to capture complex interactions and dependencies within the system by exploiting shared information across related outputs. In the context of wind turbine blade analysis, the responses of different DOFs are often governed by common load elements acting on the blade

structure. By jointly modelling these responses within a multi-task learning framework, the model can leverage shared features and patterns, leading to improved generalisation and predictive performance. For example, the numerical model presented in Section 2 uses the normal mode summation method to model the blade deformation. This approach characterises the blade response using three DOFs: $q_{B1F1}$, $q_{B1F2}$, and $q_{B1E1}$ which represent the modal coordinate of blade deformation corresponding to the first bending mode in flapwise direction, second bending mode in flapwise direction and the first bending mode in edgewise direction. Using this method, the blade deformation at any point located at a distance $x$ from the blade root at any time instance $t$ is given by Eq. 20, where $\phi$ represents the corresponding mode shape.

$$q(x,t) = \phi_{B1F1}(x)q_{B1F1}(t) + \phi_{B1F2}(x)q_{B1F2}(t) + \phi_{B1E1}(x)q_{B1E1}(t) \tag{20}$$

This deformation response can be projected into global coordinates to compute the blade tip in-plane ($q_{IP}$) and out-of-plane deformation ($q_{OP}$) using the approach presented in Ref. (Jonkman, 2003). The multi-output learning paradigm is employed to develop an LSTM model to predict the time history responses of these three DOFs. Therefore, the LSTM model capable of predicting response at these three DOFs can be used to compute multiple derived quantities such as loads, reaction and fatigue damage equivalent load without much computational overhead. In addition, the model is also extended to predict in-plane and out-of-plane blade deformations, as these quantities provide physically measurable responses for application in virtual sensing and model validation through experimental measurements. The LSTM model development and the corresponding results for these target variables are presented in the next section.

## 7 Numerical Results

In this section, numerical results are presented to demonstrate the performance of LSTM models in predicting the dynamic response of wind turbine blades under turbulent wind conditions. The methodology begins with the generation of stochastic wind fields using TurbSim, following the parameter bounds presented in Table 1 and the sampling strategy described in Section 5. These wind fields serve as input to the numerical model of the 15MW wind turbine (Section 2), which generates the training data for the LSTM model. To address the challenges posed by high-dimensional input data, while training the LSTM model, PCA and DCT are used to capture the governing spatial and temporal trends in the wind field. The RFA method is then used to select an optimal set of input features for the LSTM model.

The LSTM models developed using wind speed data alone were able to capture the time-varying mean response; however, they struggled to model fluctuations around the mean. This limitation arises because the wind turbine response is heavily influenced by the controller algorithm, which regulates the rotor speed and blade pitch angle, thereby governing the system dynamics. Predicting the response using only wind speed data would require an impractically large data set to capture the underlying dependence on controller dynamics. To address this, a multi-stage modelling approach is implemented in this study. In the first stage, LSTM models are developed to predict the controller response (rotor speed and blade pitch angle) using reduced-order representations of the wind field as input. In the second stage, the predicted control parameters from the first LSTM model are combined with the PCA and DCT components to train another LSTM model to predict the blade deformation

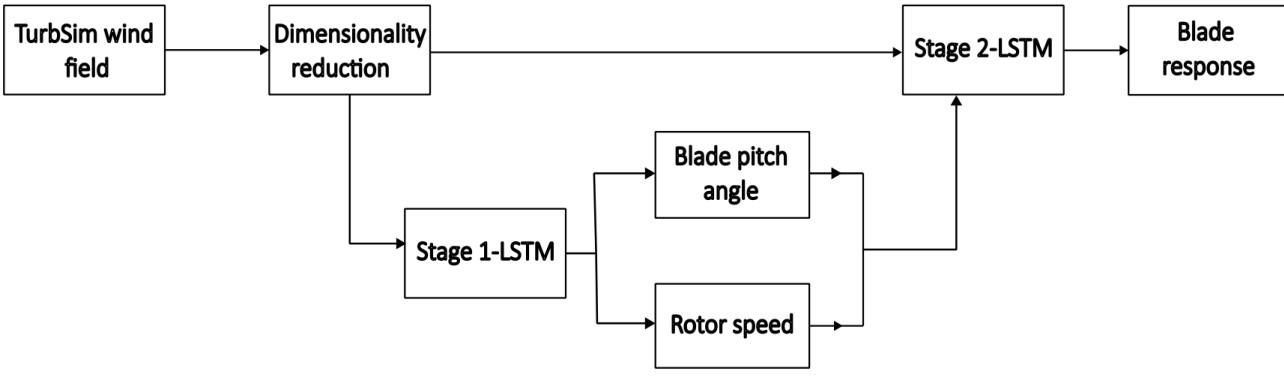

**Figure 8.** Multi-stage modelling approach for predicting blade response

response. A conceptual framework for this approach is presented in Fig. 8. This approach divides the problem into smaller, more manageable sub-tasks, consistent with the physics of the problem. The direct physical relationship between the inputs and outputs at each sub-task reduces data requirements and results in a more efficient surrogate model. The performance of each LSTM model, along with their training and results, is discussed in the next section. Wind turbine behaviour differs significantly between the above-rated and below-rated regions due to distinct controller principles. While the presented methodology can

be applied separately to each region to develop individual models, the features and model architecture may vary. To avoid information overload, this section focusses exclusively on results for the above-rated condition.

### 7.1   Predicting Control Parameters

This section presents the methodology used to develop the LSTM model to predict control parameters. The numerical results are presented, and the model performance is comprehensively analysed. The control parameters, i.e., rotor speed and blade pitch

angle, are governed by the properties of the inflow wind. These parameters are tuned using a specially designed controller to optimise the performance of a wind turbine. The controller used to regularise the rotor speed aims at achieving higher efficiency in power production in below-rated wind speeds, whereas the blade pitch angle controller aims to maintain constant power production for above-rated wind speeds. The controller algorithm introduces additional uncertainty in the response of the wind turbine. In this study, the controller algorithm is treated as a black box to develop a surrogate model. Here, an LSTM model

is developed to predict the controller response as a function of wind speed data processed through PCA and DCT algorithms. In preliminary analysis, a multi-task learning model was used to simultaneously predict the blade pitch angle and rotor speed as a function of PCA and DCT features. However, this approach failed to model both parameters simultaneously with the required level of accuracy. Alternatively, individual LTSM models developed to predict one parameter at a time were found to deliver better results. Based on these observations, the blade pitch angle and rotor speed are modelled separately through

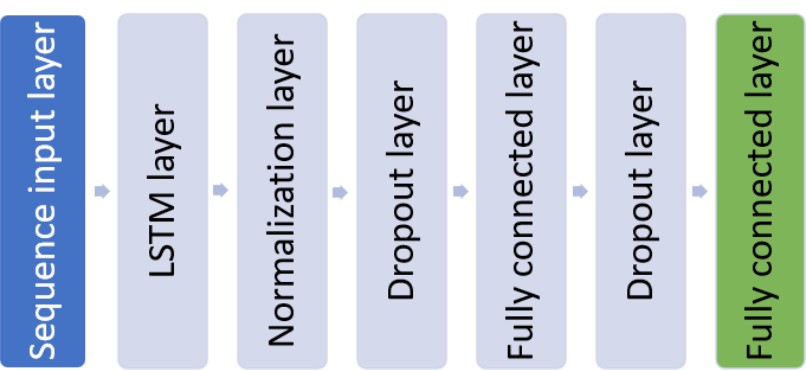

**Figure 9.** LSTM Model architecture for predicting rotor speed

individual LSTM models. In addition, it was observed that the rotor speed can be better modelled through PCA features, and
the blade pitch angle dynamics are captured more accurately using DCT features. This distinction can be attributed to the nature
of the controller responses and the type of information captured by each feature representation. Blade pitch adjustments are
typically triggered by localised and high-resolution spatial variations, such as wind shear and turbulence, which are effectively
captured by the spatial frequency decomposition offered by DCT. Rotor speed, by contrast, reflects a more integrated response
influenced by the overall wind field across both space and time, making it more sensitive to the global modes captured by PCA.
The following sections present the model architecture and its performance in predicting the individual controller response.

### 7.1.1   Predicting rotor speed response using PCA

In the preliminary analysis, the PCA approach was found to be more effective in modelling the rotor speed response. Sup-
plementing the principal component with the average rotor wind speed was found to further improve the model predictions.
This section presents the development of the LSTM model trained using principal components of wind speed data and the
rotor-averaged wind speed as input features. The model architecture is shown in Fig. 9. In this deep learning model, the se-
quence input layer inputs sequential data to the network, where the size of the sequence layer is equal to the number of input
features. The LSTM layer is used to learn long-term dependencies between time steps and extract temporal patterns from
sequential data. This layer performs additive interactions, which can help improve gradient flow over long sequences during
training (Hochreiter and Schmidhuber, 1997). Since the model should generalise well over a wide range of wind speeds, a
normalisation layer was used, which normalises a minibatch of data across all channels for each observation independently.
The use of the normalisation layer after the learning layers speeds up the training and improves the performance of the model.
A fully connected layer was then used to combine the temporal patterns and transform these extracted features. Although a
fully connected layer extracts the learnt information from the LSTM layers, it also increases the risk of overfitting. To avoid
this, a dropout layer was added to the model. A dropout layer randomly sets the input elements to zero with a given probability,

| Hyperparameter | Parameter range | Optimised values |
|---|---|---|
| LSTM layer - 1 | [1 - 100] | 81 |
| Fully connected layer - 1 | [1-100] | 64 |

**Table 2.** Summary of hyperparameter exploration for predicting controller response

which helps to avoid overfitting the training data and improves the generalisation ability of the network. In this architecture, a constant dropout probability of 10% was assumed for each dropout layer. The first fully connected layer was used to aggregate the information from the LSTM layer, while the second fully connected layer was used to represent the actual output parameters. The final fully connected layer in the model matches the number of target variables. The final fully connected layer in the LSTM model developed to predict rotor speed has only one neuron. The number of neurons in the final layer of a multi-task learning model is always greater than one. This model was trained using the Adam optimisation algorithm (Kingma and Ba, 2014). Initially, the first five principal components, which cumulatively capture 50% of the variance in the wind speed data, were selected to capture the dominant patterns in the wind speed data. This feature set was further filtered using RFA by iteratively adding features and evaluating the performance of the model to find the optimal feature set that yields the best results. During the RFA process, the rotor-averaged wind speed was considered as a fixed element in the input set. Table 2 summarises the range of hyperparameters used for exploration in this study. Hyperparameters are model parameters set prior to training that govern the model architecture and learning behaviour. Here, the number of neurons in the fully connected layer and the number of cells in the LSTM unit were chosen as the hyperparameters due to their direct influence on model capacity and performance. The parameter bounds were selected based on prior literature, preliminary experiments, and computational resource considerations. These ranges define the search space for the Bayesian optimisation technique used to tune the model. The hyperparameters are tuned using the Bayesian optimisation technique, the optimised parameter values are also presented in Table 2.

Fig. 10 shows the RMSE obtained from validation data using different principal components as input features. This figure shows that the LSTM model developed using the first principal component combined with the rotor-averaged wind speed corresponds to the lowest prediction error in the validation dataset. Here, rotor-averaged wind speed refers to the spatially averaged wind speed over the entire rotor swept area at each time instant, representing the effective inflow velocity experienced by the wind turbine rotor. Adding more input features using the RFA method did not increase the accuracy of the model predictions. To this end, the first principal component of the spatio-temporal wind, along with rotor-averaged wind speed, is used as input features to predict the rotor speed. A visual representation of the predictions of the LSTM model to the actual rotor speed across different wind fields is presented in Fig. 11, demonstrating the accuracy of the model. The overall efficiency of this approach can be contextualised by the fact that the proposed approach reduces 625 input features to only two variables capable of capturing the overall dynamics. This underscores the importance of PCA in simplifying complex datasets without sacrificing crucial information.

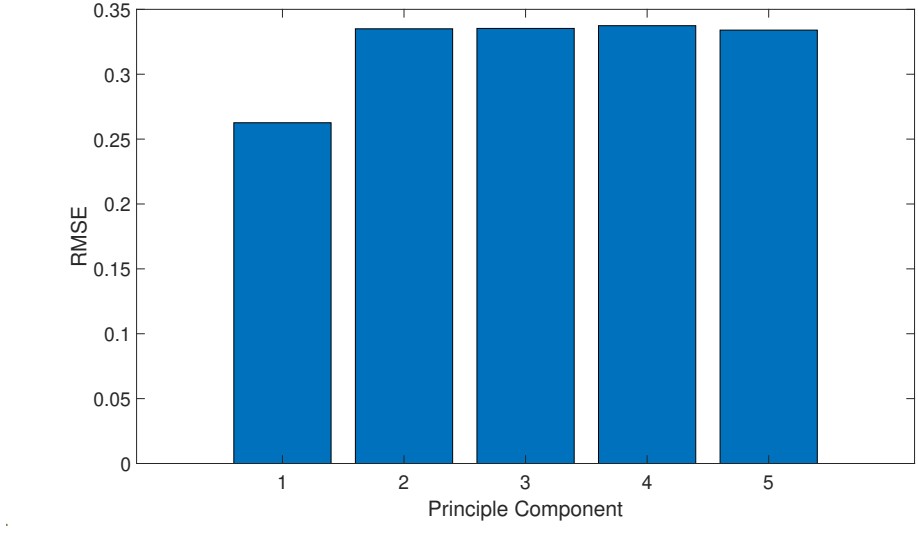

**Figure 10.** RMSE in rotor speed prediction using different principal components

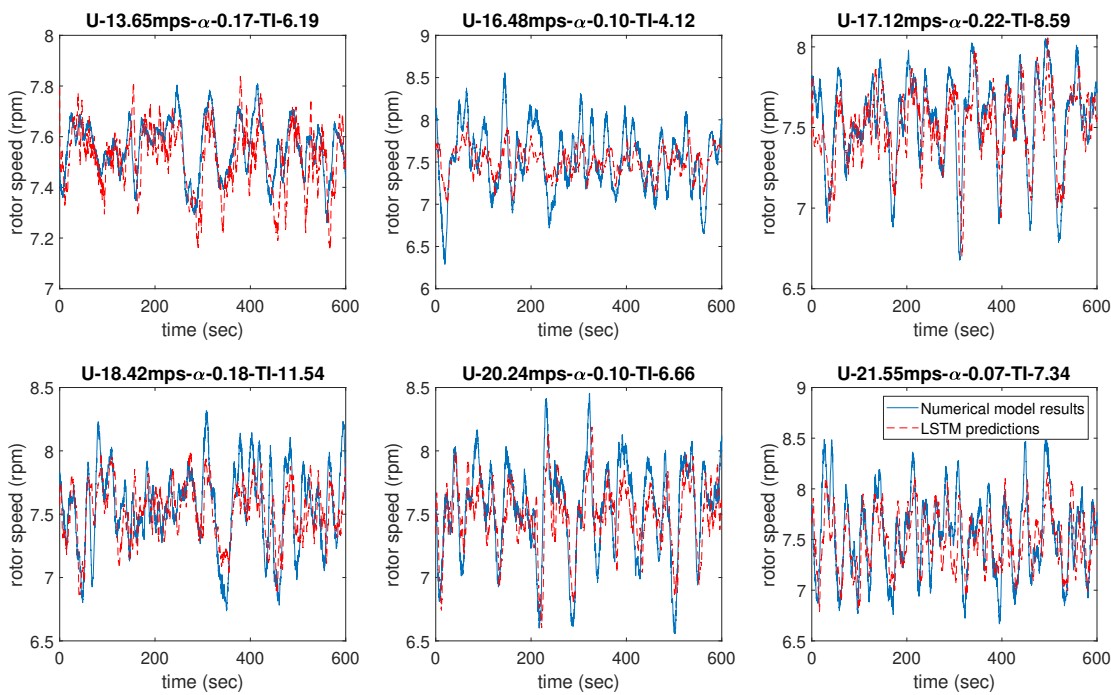

**Figure 11.** Rotor speed predictions of the LSTM model using PCA feature

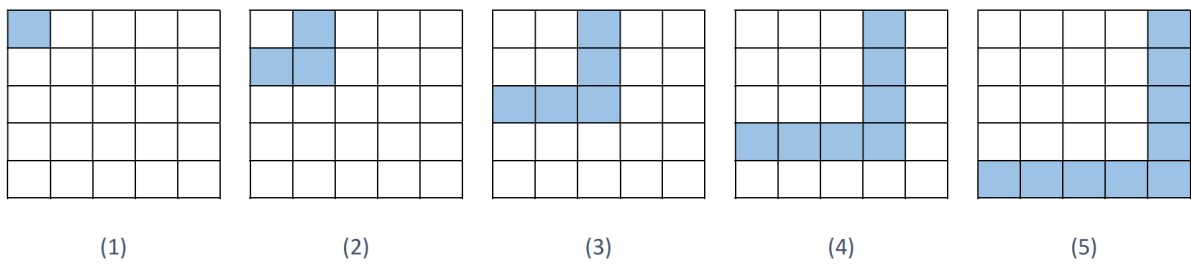

**Figure 12.** Visual representation of DCT components considered in RFA method

### 7.1.2 Predicting blade pitch response using DCT

The LSTM model used to predict the blade pitch response shares the same architecture as shown in Fig. 9. Wind speed data, transformed using DCT, served as input, and RFA was used to determine the most informative subset of DCT features for optimal prediction accuracy. Similarly to the rotor speed prediction model, the rotor-averaged wind speed acts as a fixed element of the input feature set during the RFA process of the blade pitch prediction model. In this model, the wind speed grid at each time step was treated as an image. 2D DCT is applied to this image to decompose it into different spatial frequency

components. The process of modelling wind speed data as an image and processing it through 2D DCT inherently distributes relevant information across the off-diagonal terms of the output matrix. As such, the off-diagonal terms in the DCT matrix share mutual information about spatial frequencies in the horizontal and vertical directions. To capture this information, the DCT components were grouped to form feature subsets, each representing an incremental higher frequency subset. A mathematical definition of a feature subset $z_i$ is given by Eq. 21

$$z_i = Z_{DCT}n \times n(1:i, 1:i) - Z_{DCT}n \times n(1:i-1, 1:i-1) \forall 1 \leq i \leq n \tag{21}$$

A visual representation of component groups of the DCT matrix is shown in Fig. 12. This figure illustrates the grouping of 2D DCT components for a $5 \times 5$, as described in Eq. 21. Each group $z_i$ represents an incremental subset of spatial frequency components, starting from the lowest frequencies (top-left corner of the matrix) and progressively including higher frequencies. This grouping captures the mutual information shared across the off-diagonal terms of the DCT matrix, which encode spatial

frequency information in both horizontal and vertical directions. The visual representation in the figure highlights how each subset of features $z_i$ is constructed taking the difference between successive submatrices of the DCT output. Fig. 13, presents a representative $5 \times 5$ matrix of a wind field where the spatial frequency in the horizontal direction increases from left to right, and the frequency in the vertical direction increases from top to bottom. In each successive iteration of the RFA method, the input features in Fig. 13 corresponding to the highlighted components of the matrix in Fig. 12 are selected. Using this feature

definition along with the RFA scheme, the LSTM model was trained to predict the blade pitch response.

The initial $5 \times 5$ submatrix of the $25 \times 25$ matrix obtained through 2D DCT was selected as the dominant subset of input features, as it represents the lower frequency patterns which can more effectively capture general trends in the wind speed

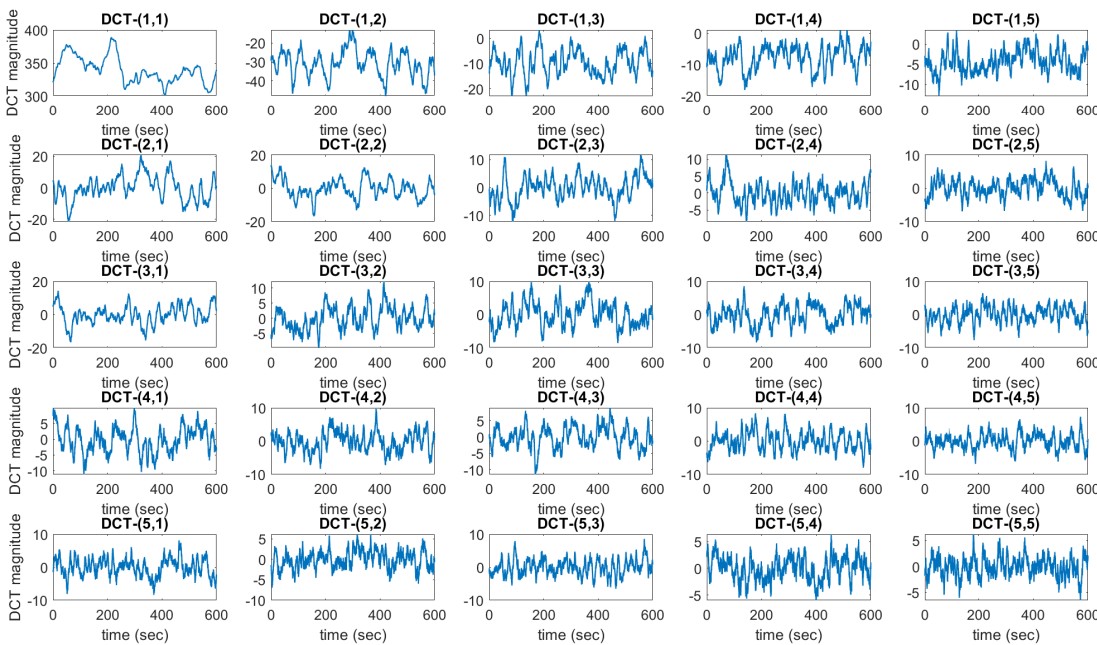

**Figure 13.** $5 \times 5$ submatrix of DCT representation of wind field

data. Fig. 14 presents the RMSE obtained for the LSTM model trained on each group of DCT components (presented in Fig. 12), demonstrating that the first component, representing the lowest frequency patterns extracted by the DCT, achieves

the best performance in predicting the pitch angle of the blade. This signifies that low-frequency fluctuations in wind speed play a dominant role in driving the blade pitch response. The findings of the RFA showing the best predictive capacity of the LSTM under the low-frequency component can be explained using control theory. The control system governing blade pitch angle is designed to optimise turbine performance and stability by filtering out high-frequency disturbances and adjusting the blade pitch angle in a more deliberate and controlled manner. This intentional smoothing of blade pitch variations serves to

mitigate the impact of sudden changes in wind speed and maintain the overall stability and efficiency of turbine operation. Fig. 15 further reinforces this observation by comparing the actual blade pitch response to the LSTM model predictions for different wind fields. The proposed model efficiently captures the overall trends in blade pitch response across different mean wind speeds, highlighting the generalisability of the LSTM model. The predictions of rotor speed and blade pitch angle are combined with the PCA and DCT data to predict the deformation response at the blade tip. In this stage, the rotor speed was

transformed into the angular position of the blade so that it could more effectively capture the rotation frequency.

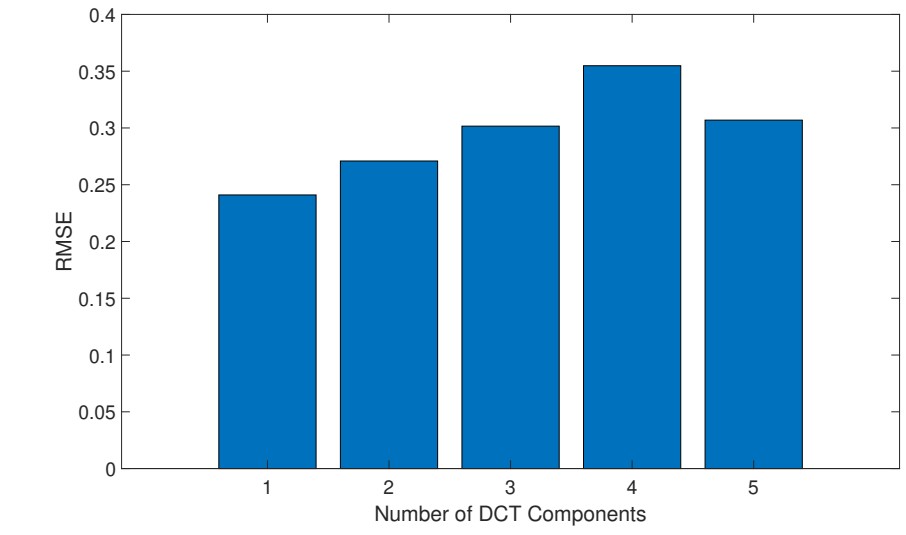

**Figure 14.** RMSE in blade pitch prediction using different DCT components

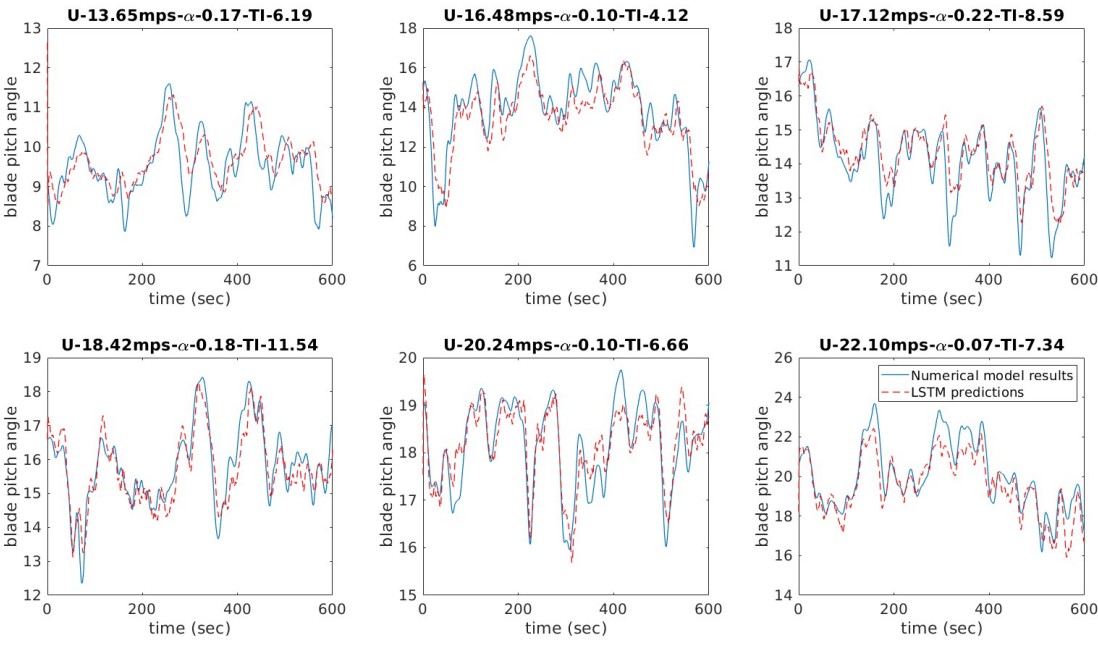

**Figure 15.** Blade pitch angle predictions of the LSTM model using DCT features

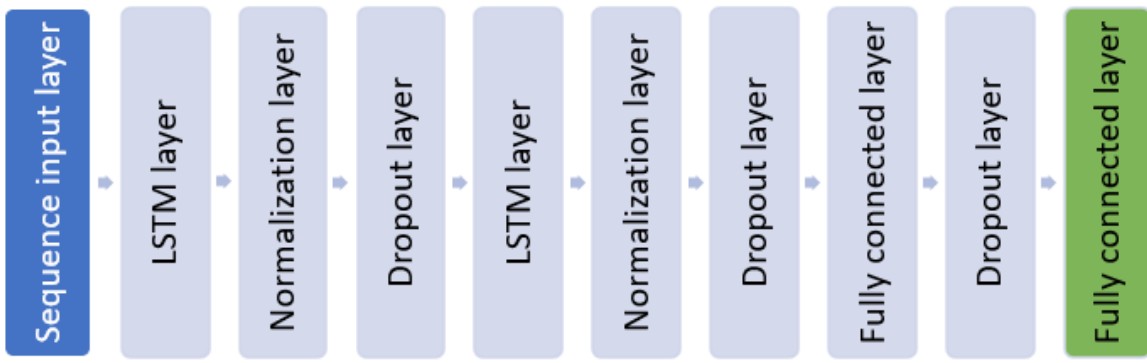

**Figure 16.** LSTM Model architecture for predicting blade response

## 7.2 Predicting blade response

In this section, a two-layer LSTM model architecture was implemented to predict the deformation response of the wind turbine blade. To explore the effectiveness of the PCA and DCT methods, two sets of input features are considered:

1. Principal components of the wind field, combined with the angular position of the blade and the blade pitch angle predictions of the LSTM models described in Sections 7.1.1 and 7.1.2.

2. Coefficients of the discrete cosine transform of the wind field, combined with predictions of the pitch angle and the angular position of the blade from the LSTM models described in Sections 7.1.1 and 7.1.2.

The results of these two approaches were compared to assess the capabilities of PCA and DCT in the context of blade response prediction. A multi-task learning approach was employed to simultaneously predict the blade response. Initially, the model was trained to predict in-plane and out-of-plane deformations, which was subsequently extended to predict the response of individual blade DOFs in the flap- and edge-wise directions. Matlab® codes used for model training and validation are available at Baisthakur (2024).

### 7.2.1 Blade response prediction with PCA features

Following the approach outlined in the previous sections, this section presents the model developed to predict the blade response and the corresponding results. The architecture of the model for the prediction of blade response is detailed in Fig. 16. The LSTM architecture is designed through an iterative process, and hyperparameter optimisation is used to identify the most optimal parameters for various layers. The number of hidden units in the learning layers was used as a hyperparameter in this study. The performance of the model is evaluated using Normalised Root Mean Square Error (NRMSE) between the predictions

| Hyperparameter | Parameter range | Optimised values |
|---|---|---|
| LSTM layer - 1 | [100 - 200] | 138 |
| LSTM layer - 2 | [50 -100] | 63 |
| Fully connected layer - 1 | [1-50] | 44 |

**Table 3.** Summary of hyperparameter exploration to optimise the number of hidden units in learning layers

in the validation data and the actual results. NRMSE is used as an error metric because it normalises the RMSE by the range of the observed data, making it suitable for comparing model performance across signals with different magnitudes, such as in-plane and out-of-plane deformations. The use of NRMSE ensures interpretability and fairness in evaluating the model's accuracy, as it accounts for variations in signal scales, which is particularly important when analysing multi-task learning outcomes for wind turbine blade dynamics. Table 3 summarises the hyperparameter exploration. A progressively smaller range is used for the bounds of hyperparameters, as consistently decreasing the number of hidden units was found to deliver better results. Using the angular position of the blade and the pitch angle as constant features (i.e., features permanently included in the input set throughout the feature selection process), the principal components of the wind speed data were selected using the RFA algorithm. In this context, the RFA algorithm does not start from an empty set but always includes these two features as part of the input, which are governed by the physics of the problem. This approach enables accurate predictions of both in-plane and out-of-plane blade deformations (Fig. 17). In particular, the second principal component, when paired with the angular position and pitch angle of the blade, yielded the best performance. This suggests that the second principal component likely captures patterns more relevant to predicting blade deformations than the first principal component, despite the fact that the first component captures the maximum variance in the data set. These findings further reinforce the importance of using feature selection approaches in obtaining a simplified model representation with a small set of the most informative features. This method is further extended to predict the response at individual blade DOFs, which are the most fundamental quantities in numerical modelling, and the corresponding results are presented in Fig. 18 and Fig. 19. Using the fifth principal component as the input feature in conjunction with the blade angular position and pitch angle produced the lowest RMSE to predict the response at individual blade DOFs.

Here, it can be seen that a higher level of accuracy was obtained in the prediction of DOFs $q_{B1F1}$ and $q_{B1E1}$ as compared to $q_{B1F2}$. This can be attributed to the higher frequency of the mode shape for $q_{B1F2}$. Furthermore, the absolute magnitude of $q_{B1F2}$ shows that this DOF has a lower impact on total deformation compared to the response of $q_{B1F1}$. However, it is evident from Fig. 18 and Fig. 19 that a good approximation in predicting all the DOFs is achieved by capturing the governing dynamics in all the parameters.

### 7.2.2 Blade response prediction with DCT features

Building upon the PCA approach, the use of DCT features for the prediction of the blade response is explored in this section. The LSTM model follows the same model architecture presented in the previous section (Fig. 16). Following the methodology

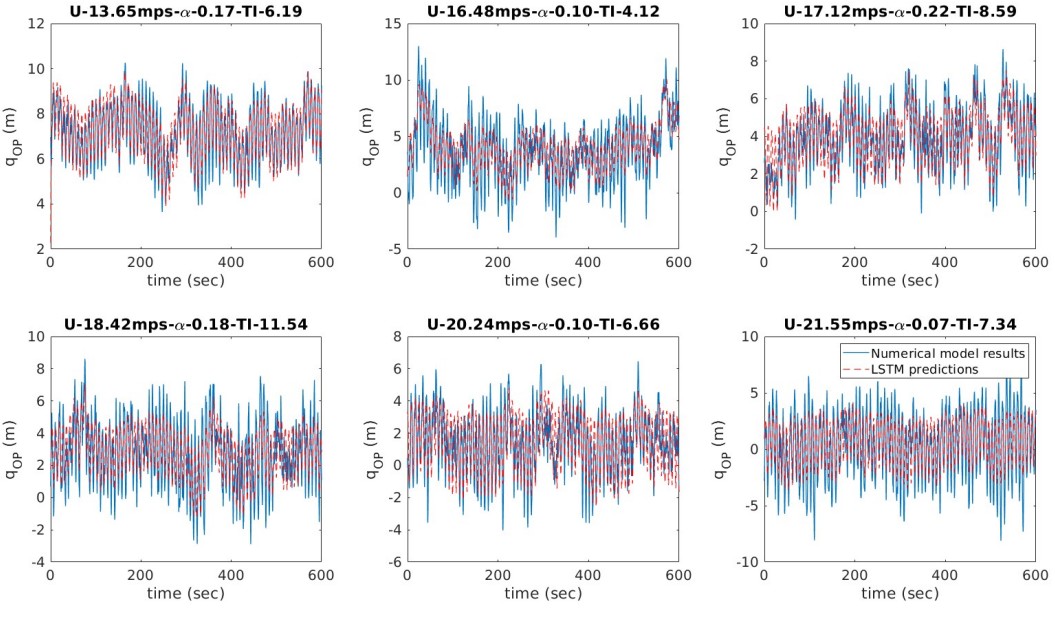

(a) Blade out of plane deformation response

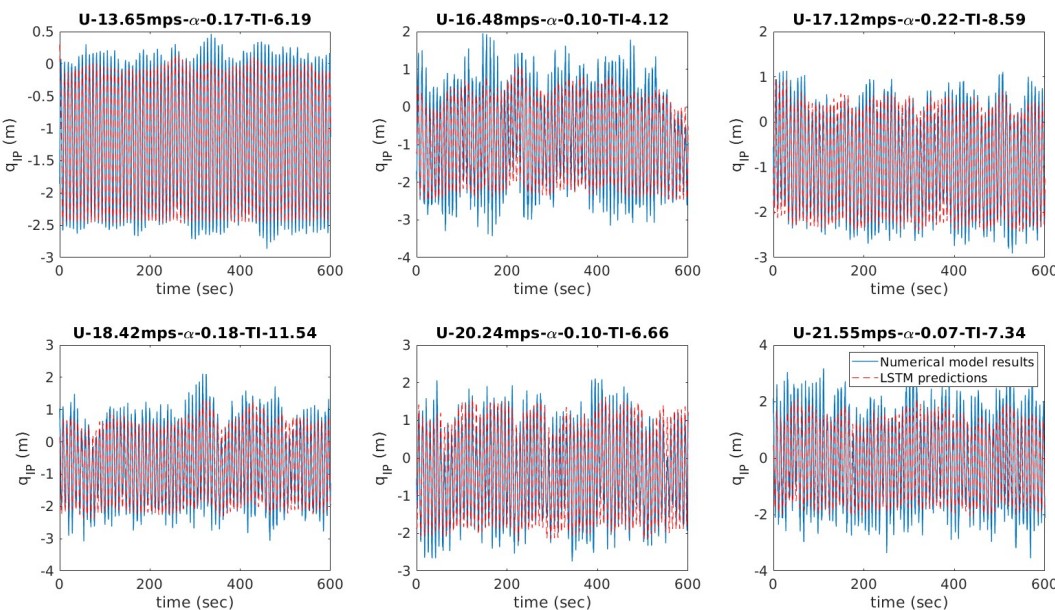

(b) Blade in-plane deformation response

**Figure 17.** Multi-task learning LSTM Model predictions of blade deformations using PCA features

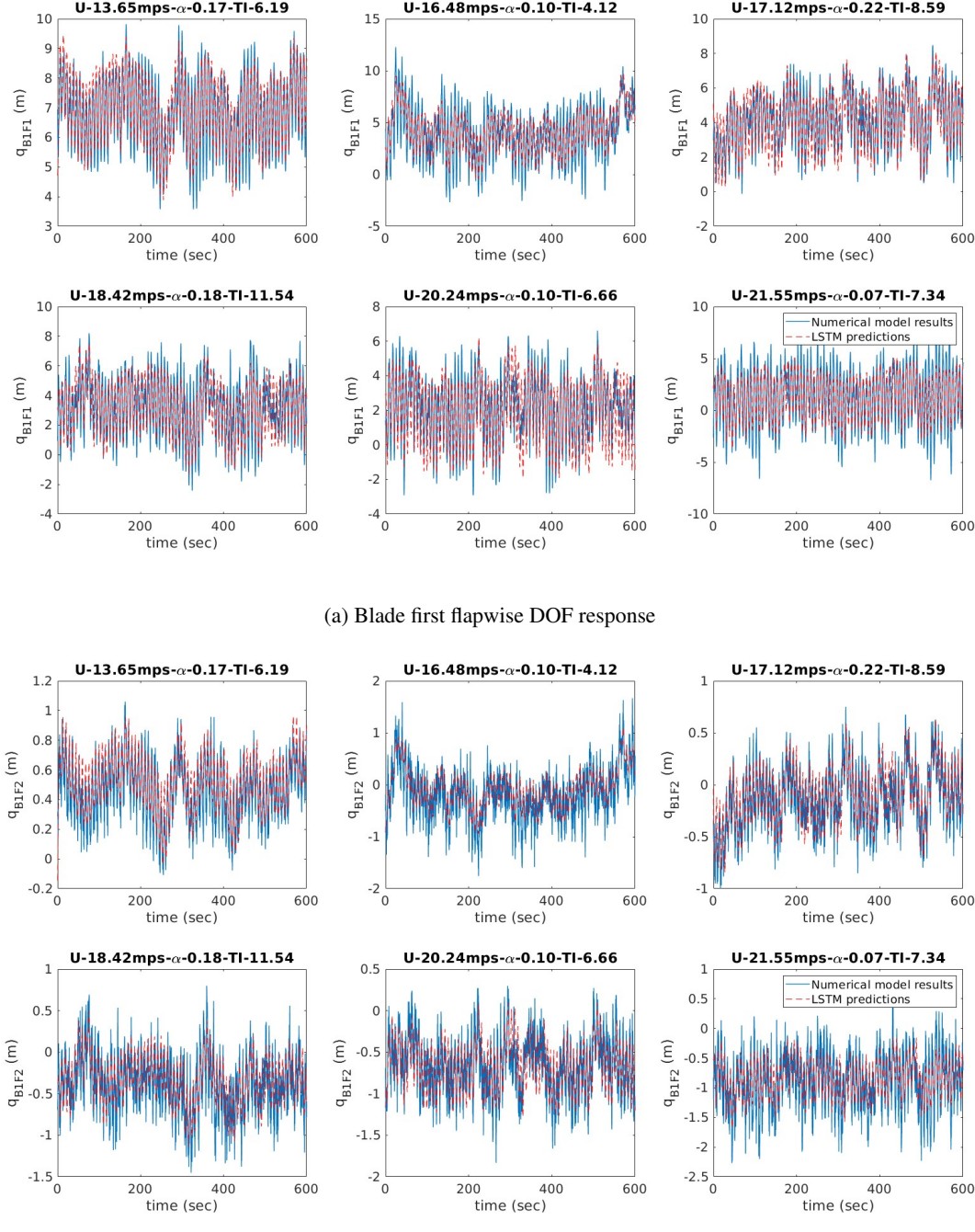

(a) Blade first flapwise DOF response

(b) Blade second flapwise DOF response

**Figure 18.** Multi-task learning LSTM Model predictions of blade flapwise DOFs using PCA features

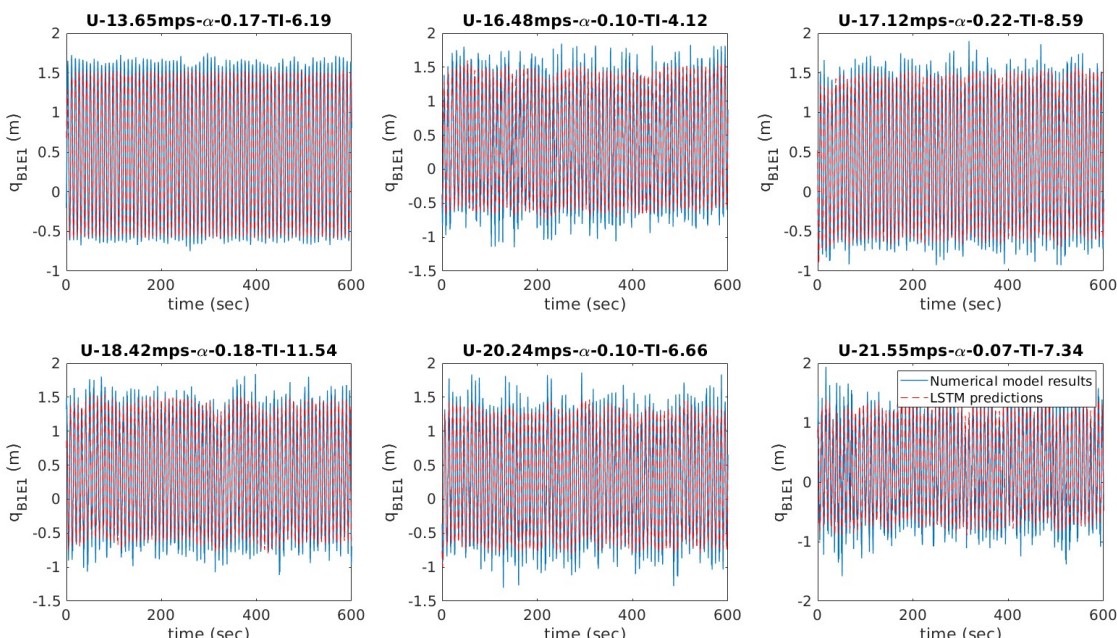

**Figure 19.** Multi-task learning LSTM Model predictions of blade edgewise DOF using PCA features

| Output Parameter | NRMSE | |
|---|---|---|
| | PCA | DCT |
| Blade IP Deformation | 0.074 | 0.047 |
| Blade Oop Deformation | 0.083 | 0.059 |
| Blade DOF -$q_{B1F1}$ | 0.092 | 0.081 |
| Blade DOF -$q_{B1F2}$ | 0.112 | 0.102 |
| Blade DOF -$q_{B1E1}$ | 0.073 | 0.065 |

**Table 4.** Comparison of LSTM Model performance trained using PCA and DCT features

adopted in Section 7.2.1, the model architecture was trained using DCT features. Fig. 20, Fig. 21, and Fig. 22 demonstrate the precision of DCT in predicting the blade response, capturing the dynamics of the target variable. The best performance for blade response prediction was obtained using the second group of DCT components presented in Fig. 12 as input features, while the blade DOF predictions achieve higher accuracy under the combination of the second and third groups of DCT components.

Table 4 provides a comparison of the NRMSE obtained for models developed using the PCA and DCT features, highlighting DCT's superior performance in predicting the blade response. These results highlight that the LSTM model can efficiently predict the dynamic response of wind turbine blades. The accuracy of the predictions also depends on the dimensionality

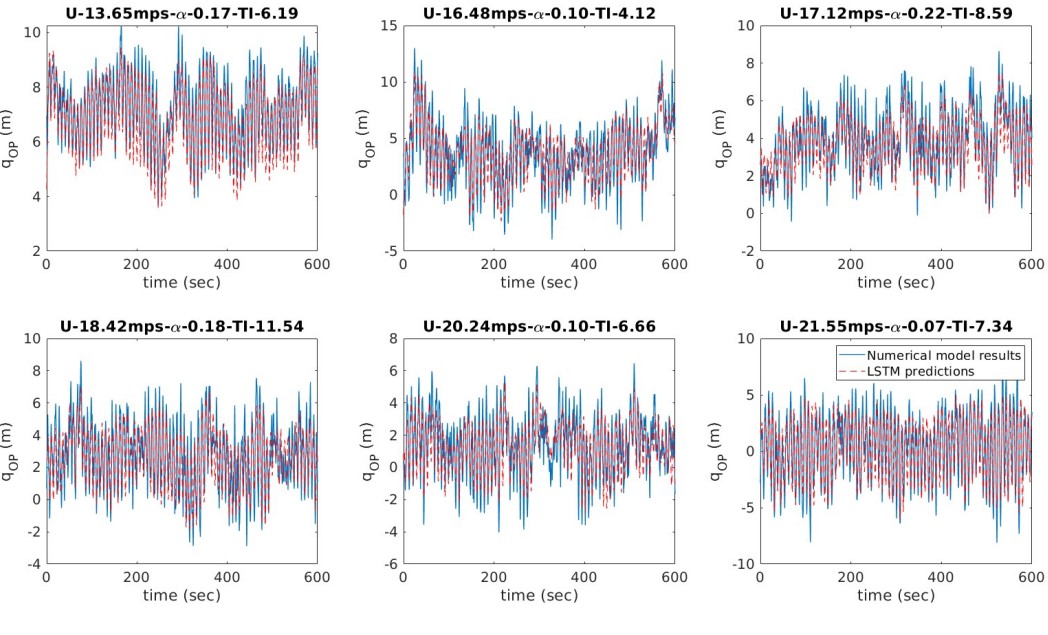

(a) Blade out of plane deformation response

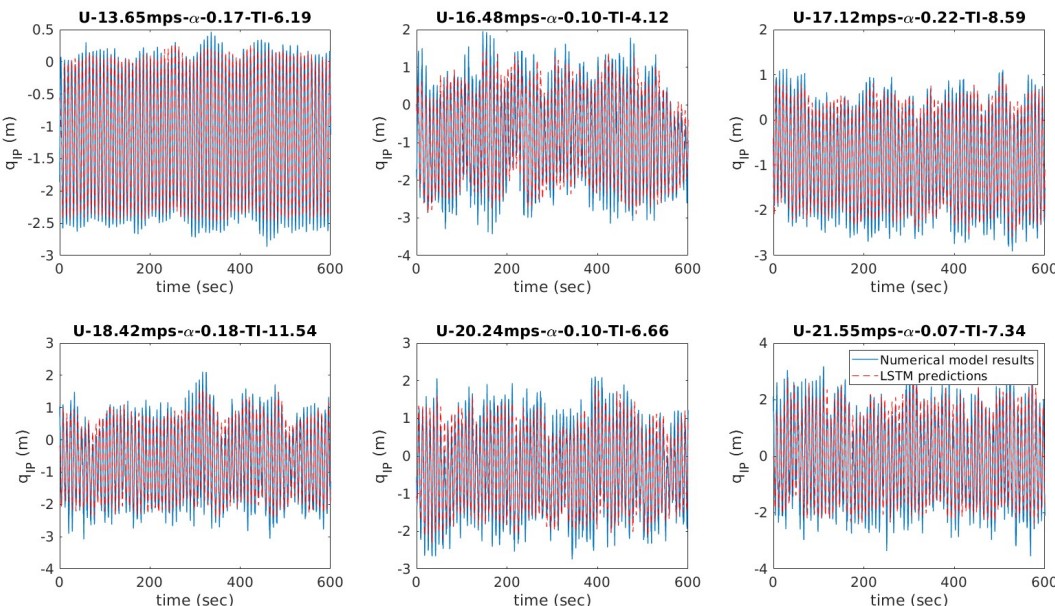

(b) Blade in-plane deformation response

**Figure 20.** Multi-task learning LSTM Model prediction of blade deformations using DCT features

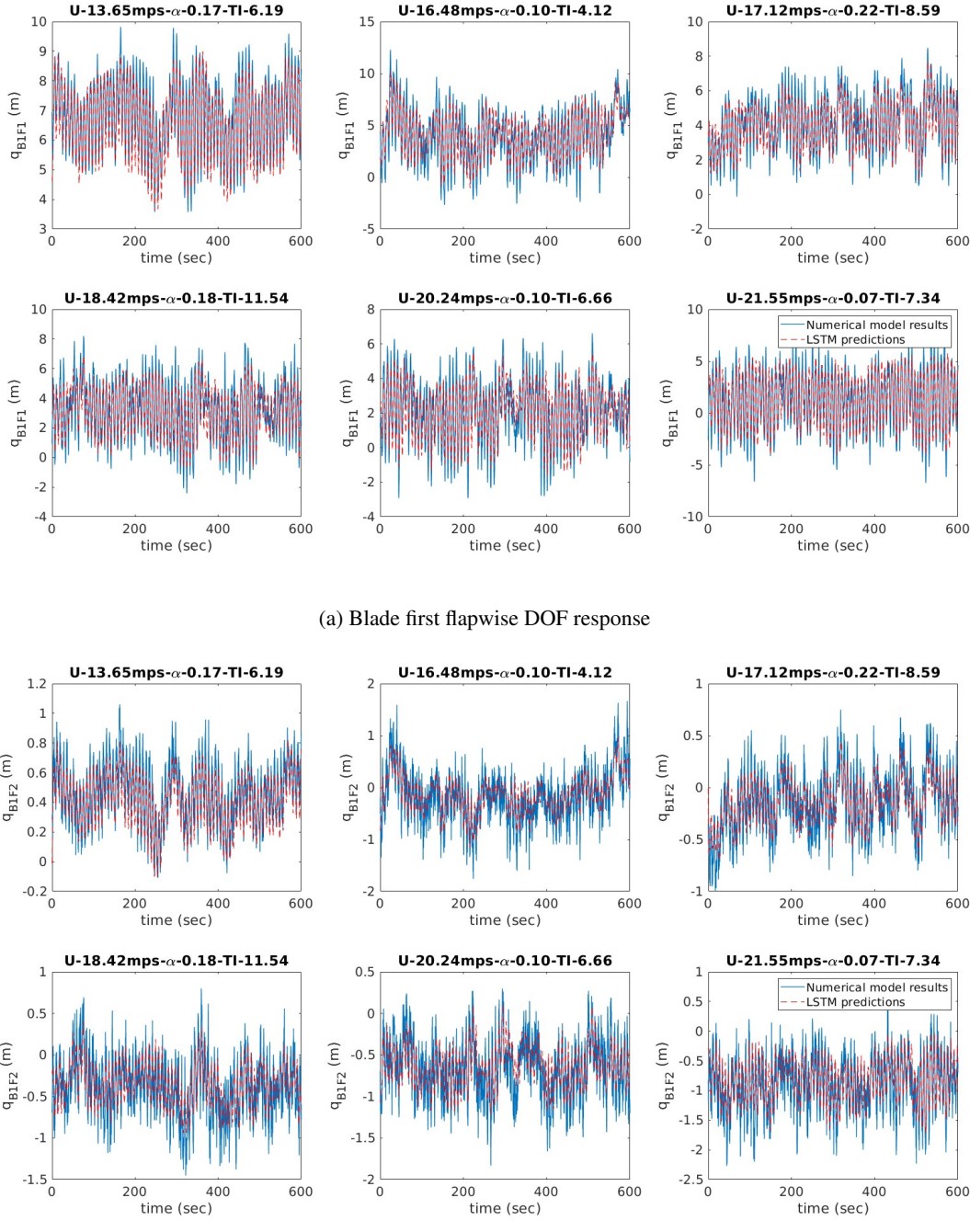

(a) Blade first flapwise DOF response

(b) Blade second flapwise DOF response

**Figure 21.** Multi-task learning LSTM Model prediction of blade flapwise DOFs using DCT features

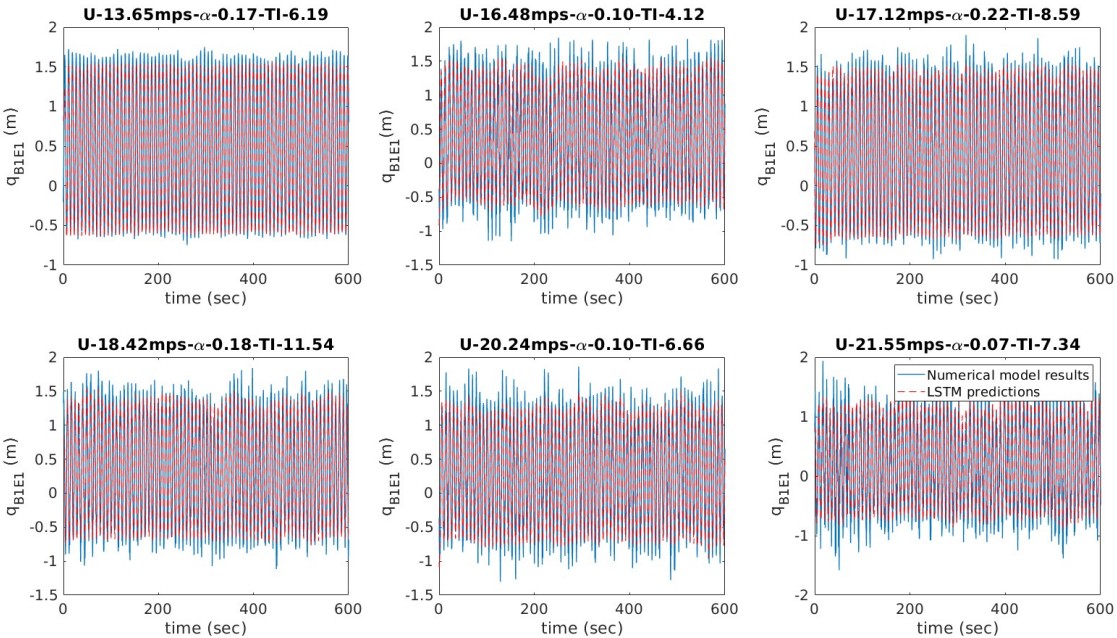

**Figure 22.** Multi-task learning LSTM Model predictions of blade edgewise DOFs using DCT features

reduction algorithm and the features chosen for the development of the model. The accuracy achieved in the intermediate stage of the multi-stage modelling approach also significantly impacts the final model predictions. These results also demonstrate
that DCT, by focussing on spatial frequency patterns, delivers better results in predicting the blade responses. While Temporal Convolutional Networks (TCNs) have been successfully used to predict damage equivalent loads (DELs) using wind speed data (Haghi and Crawford, 2023), direct parallels cannot be drawn between these studies to compare model performance against LSTM due to differences in objectives and scope. For example, the referenced study uses TCNs to predict a single variable (DEL) for a 5MW wind turbine, whereas this work focusses on predicting the time-history response of multiple fundamental
quantities for a 15MW turbine. In the realm of machine learning modelling, the suitability of a particular model depends on several factors, including data availability, computational resources, model complexity, and the specific requirements of the application. Only when these fundamental objectives are the same, a comprehensive comparison of various surrogate modelling techniques, including TCNs and LSTMs, would be an interesting direction for future research. To this end, while TCNs have shown promise in certain tasks, LSTMs are found to be able to capture the complex temporal dependencies inherent in wind
turbine dynamics, as demonstrated in this study. The next section focusses on analysing the computational advantage of the developed surrogate models.

### 7.3 Computational advantage of surrogate approach

The need to reduce the computational cost in predicting the dynamic response of a wind turbine is at the heart of the surrogate model developed in this study. Having established the accuracy of the developed surrogate model, this section focusses on the computational advantages of using these models. A direct comparison of execution times was conducted to quantify the extent of the computational efficiency of these models. Both the numerical model and the surrogate model were used to predict blade response for a 600-second turbulent wind inflow generated using TurbSim. To ensure a fair comparison, only the essential blade DOFs and generator azimuth angle were activated within the numerical model. The surrogate modelling approach developed in this study offers a significant computational advantage in predicting blade response. The execution time comparison shows a 75-fold speed improvement using the surrogate model compared to the corresponding numerical model. This comparison highlights the computational efficiency of the developed surrogate models. The computational advantage allows for conducting a large number of simulations required for site-specific performance analysis. Furthermore, the ability of the surrogate model to quickly predict blade deformations under varying wind and control inputs makes it a useful tool for comprehensive fatigue analysis of wind turbine blades. While quantifying the computational gains, all the simulations were performed on a system with an 8-core Intel Xeon CPU with a clock speed of 3.8GHz, using 32GB RAM and running on Microsoft Windows 10 Pro. It is noted that the computational advantage reported here pertains to the prediction phase of the surrogate model. Although the data generation, model training, and feature selection incur a one-time computational cost, this overhead is significantly offset when the model is used for large-scale simulation tasks such as fatigue analysis or design exploration.

## 8 Summary and Conclusion

In this manuscript, an approach that combines multi-stage modelling and multi-task learning with dimensionality reduction techniques and a feature selection algorithm was presented. This combined method aims to enhance the efficiency of developing LSTM models for predicting blade response. Based on the investigation performed in this paper, the following conclusions can be drawn:

– Dimensionality Reduction: The effectiveness of PCA and DCT in simplifying wind field data while retaining crucial information for prediction tasks was demonstrated. Both PCA and DCT, particularly when combined with recursive feature addition, helped in achieving efficient model configuration and improved prediction accuracy.

– Rotor Speed and Blade Pitch Prediction: LSTM models were developed for rotor speed prediction (using PCA features) and blade pitch prediction (using DCT features). These models achieved varying degrees of accuracy in capturing the dynamics of their respective target parameters across different uncertainty levels in wind conditions. Accurate rotor speed and blade pitch information were identified as critical parameters for subsequent blade response prediction.

– Blade Response Prediction: A multi-stage modelling approach was employed, where predicted control parameters were fed into an LSTM model to predict blade deformations and DOF responses. Models trained using DCT features showed

higher accuracy as compared to PCA features for this task, indicating DCT's ability to capture spatial frequency patterns driving blade dynamics.

– Practical applications: The LSTM model presented in this paper is trained using input-output data only. This approach has potential applications in design feasibility studies for those models where the exact model configuration is not available due to intellectual property concerns. Further, due to the low computational cost, these models can be used within a model predictive control framework for regulating the performance.

## 9   Limitations and Future Work

The surrogate modelling framework presented in this study provides a computationally efficient means of predicting wind turbine blade response using LSTM networks, informed by recursive feature selection tailored to aerodynamic and structural dynamics. While the results demonstrate good predictive performance within the scope of the current study, several limitations and directions for future work remain.

    A primary limitation lies in the use of synthetic wind input fields generated under controlled assumptions. Although the wind
fields used for model training span a realistic range of turbulence intensities and spectral properties, they do not capture full-scale atmospheric variability. Incorporating real-field data or high-fidelity meteorological simulations could improve robustness and extend applicability. Additionally, the model is currently trained for a specific turbine configuration and structural model. While the learning architecture is generalisable in principle, transferability to different turbine sizes, control strategies, or structural layouts requires careful re-validation. Future work could explore transfer learning strategies to reduce retraining
effort across different turbine designs. While this study demonstrates the ability of the surrogate model to accurately capture the dynamic response of wind turbine blades, its application to downstream tasks such as fatigue analysis may introduce additional sources of uncertainty. Since fatigue assessments are sensitive to the accumulation of cyclic loading effects over time, even small prediction errors in the dynamic response, particularly in phase or amplitude, could influence the accuracy of damage estimates. It is therefore essential to evaluate whether such discrepancies consistently amplify, attenuate, or statistically
cancel out over extended simulations. Future work will focus on quantifying this uncertainty to establish confidence bounds when using surrogate models for long-term performance and reliability assessments. Finally, although computational gains during inference are significant, the training phase remains time- and resource-intensive.

*Code and data availability.*   The code and data presented in this study are available on request from the corresponding author. The data are not publicly available because it also forms part of an ongoing study.

*Author contributions.* Shubham Baisthakur: Writing – review & editing, Writing – original draft, Software, Methodology, Investigation, Formal analysis, Data curation, Conceptualization. Breiffni Fitzgerald: Writing – review & editing, Supervision, Resources, Project administration, Methodology, Investigation, Funding acquisition, Conceptualization.

*Competing interests.* There are no competing interests in the work presented.

*Acknowledgements.* This work is supported by Science Foundation Ireland (SFI) grant no. 20/FFP-P/8702.

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
