# Peer review of "Multi-task Learning Long Short-term Memory Model to Emulate Wind Turbine Blade Dynamics"

_Wind Energy Science, 2024_

## Referee Comment (RC2)

[revised manuscript text omitted]

(a) Blade first flapwise DOF response

(b) Blade second flapwise DOF response

[Figure]

(c) Blade first edgewise DOF response

**Figure 15.** Multi-task learning LSTM Model predictions of blade DOFs using PCA features

[Figure]

[Figure]

| Output Parameter | RMSE | |
|---|---|---|
| | PCA | DCT |
| Blade IP Deformation | 0.518 | 0.315 |
| Blade Oop Deformation | 0.595 | 0.378 |
| Blade DOF -$q_{B1F1}$ | 0.602 | 0.565 |
| Blade DOF -$q_{B1F2}$ | 0.665 | 0.637 |
| Blade DOF -$q_{B1E1}$ | 0.469 | 0.454 |

**Table 5.** Comparison of LSTM Model performance trained using PCA and DCT features

550    prediction was obtained using the second group of DCT components presented in Fig. 10 as input features, while the blade DOF predictions achieve higher accuracy under the combination of the second and third groups of DCT components.

Table 5 provides a comparison of RMSE obtained for models developed using PCA and DCT features, highlighting DCT's superior performance in predicting the blade response. These results highlights that LSTM model can accurately predict the dynamic response of wind turbine blades. The accuracy of predictions also depends on the dimensionality reduction algorithm

555    and the features chosen for the model development. The accuracy achieved in the intermediate stage of the multi-stage modelling approach also significantly impacts the final model predictions. These results also demonstrate that DCT, by focusing on spatial frequency patterns, delivers better results in predicting the blade responses. The next section focuses on analyzing the computational advantage of the developed surrogate models.

**8.3 Computational advantage of surrogate approach**

560    The need to reduce the computational cost in predicting the dynamic response of a wind turbine is at the heart of the surrogate model developed in this study. Having established the accuracy of the developed surrogate model, this section focuses on the computational advantages of using these models. A direct comparison of execution times was conducted to quantify the extent of the computational efficiency of these models. Both the numerical model and the surrogate model were used to predict blade response for a 600-second turbulent wind inflow generated using TurbSim. To ensure a fair comparison, only the essential

565    blade DOFs and generator azimuth angle were activated within the MBD model. The surrogate modelling approach developed in this study offers a significant computational advantage in predicting blade response. The execution time comparison shows a 75-fold speed improvement using the surrogate model compared to the corresponding MBD model. This comparison highlights the computational efficiency of the developed surrogate models. The computational advantage allows for conducting a large number of simulations required for site-specific performance analysis. Furthermore, the ability of the surrogate model to

570    quickly predict blade deformations and loads under varying wind and control inputs makes it a useful tool for comprehensive fatigue analysis of wind turbine blades. While quantifying the computational gains, all the simulations were performed on a system with an 8-core Intel Xeon CPU with a clock speed of 3.8GHz using 32GB RAM and running on Microsoft Windows 10 Pro.

[Figure]

[Figure]

(a) Blade out of plane deflection response

[Figure]

(b) Blade in-plane deflection response

**Figure 16.** Multi-task learning LSTM Model prediction of blade deflections using DCT features

[Figure]

[Figure]

(a) Blade first flapwise DOF response

(b) Blade second flapwise DOF response

[revised manuscript text omitted]

---

## Author Response (AR1)

**Response to Reviewer Comments**

Shubham Baisthakur and Breiffni Fitzgerald

March 13, 2025

We sincerely thank the reviewer for their thorough review of our manuscript titled *"Multi-task Learning Long Short-term Memory Model to Emulate Wind Turbine Blade Dynamics"* and for providing valuable feedback. The reviewer's comments have greatly contributed to improving the overall quality of this paper.

In response to the reviewer's comments, this document lists each comment in *italic text*, followed by our responses in standard text. In addition, a separate PDF file is attached, addressing the detailed comments provided by the second reviewer.

**Reviewer 1**

*The article presents a method of developing LSTM models for predicting blade response using multi-stage modelling and multi-task learning with dimensionality reduction techniques. Overall, the article is well-written, presenting a novel and relevant contribution in data driven methods. There are several minor aspects of the article which could be modified to improve its quality.*
* * *
***Q 1.1*** *Article can be made more concise - Consider shortening section 2 – not necessary to explain how TurbSim works, possible to reorganise by removing section 2 entirely and including the important parts as part of section 6*

**Reply**: We thank the reviewer for this suggestion. In the revised manuscript, the content of Section 2 (Generation of stochastic wind field) has been merged with Section 6 (Data Generation: Selection of Input Variables), and only the relevant information for this study has been retained.
* * *
***Q 1.2*** *Sufficient to simply state that IEA-15MW reference turbine is used – Figure 2 and Table 1 not necessary*

**Reply**: Figure 2 and Table 1 have been removed from the manuscript to improve conciseness. For further details on the parameters of the 15 MW wind turbine, readers are directed to the original document, which has been cited in the revised manuscript.
* * *
***Q 1.3*** *In Section 1 or 5 – should include a justification on why LSTM (or RNNs in general) is used in this work, as opposed to conventional Neural Networks*

**Reply**: Thank you for your suggestion. A brief discussion has been added to Section 5, explaining the rationale for choosing LSTM over conventional neural networks. This justification has been included to provide clarity on the choice of algorithm.
* * *
***Q 1.4*** *Table 3 – Optimized value for Fully connected layer is at upper limit – why is range not expanded to ensure it is actually optimal?*

**Reply**: We thank the reviewer for their insightful suggestion. In response, we have expanded the optimisation bounds for the fully connected layer to ensure that the obtained solution is indeed optimal. The updated results, presented in the revised manuscript, show different parameter values; however, these changes did not significantly impact the model performance. Since the overall predictive accuracy and behaviour of the

model remain consistent, we have retained the original figures in the manuscript to maintain continuity with the discussion and results presented.
* * *
**Q 1.5** *Table 5 – Between RMSE and signals shown in Figures 14 to 17, it is clear that LSTM model is accurate. However, RMSE may not necessarily be the best metric to measure model performance, as it does not take into account the variation of the signal (e.g. oop deflection has much larger magnitudes than ip deflection). Consider the use of other metrics such as Variance Accounted For or Confidence Index.*

**Reply**: We thank the reviewer for their suggestion. To better account for signal variation, we have replaced RMSE with Normalised Root Mean Square Error (NRMSE) in the revised manuscript. NRMSE normalises errors by the data range, making it more suitable for comparing signals with different magnitudes, such as oop and ip deflections. This change aligns with the reviewer's feedback and is also consistent with the suggestions of the second reviewer.
* * *
**Q 1.6** *Figure 10 – Unclear on what this figure represents – some additional explanation in caption may be helpful*

**Reply**: We thank the reviewer for this suggestion. To improve the clarity of Figure 10, additional details have been added to the manuscript, explaining the grouping of 2D DCT components and their significance in capturing spatial frequency information. In addition, an additional figure has been included showing a representative example of DCT components, to aid the reader in understanding how feature subsets are constructed and utilised in the model.
* * *
**Q 1.7** *Example LSTM predictions (Figures 9,12,14 to 17) show results at TI=11.54 – however, range from Table 2 shows that max TI should be 7.629*

**Reply**: The bounds provided in Table 2 correspond to the variance in wind speed $\sigma_u$. Since the turbulence intensity (TI) is defined as $TI = \frac{\sigma_u}{U}$ where $U$ represents the mean wind speed. As such, TI values for above-rated wind speeds can achieve values higher than 7.629%. These changes are reflected in the modified manuscript.
* * *
**Q 1.8** *Example LSTM predictions (Figures 9,12,14 to 17) only show U = 13.65m/s upwards and TI=6.19 to 11.54 – may be interesting to see predictions at lower speeds or higher TI (if possible)*

**Reply**: We thank the reviewer for their suggestion. While the manuscript focuses on the operational range of the turbine, we have generated an additional figure showing LSTM predictions at higher turbulence intensities (TI) to address the reviewer's interest. This figure is provided as supplementary material for the reviewer's consideration. We believe this demonstrates the model's capability under extended conditions without altering the main results presented in the manuscript.

**Reviewer 2**

*The article presents a method to a) a simple dynamic model of wind turbine b) unsteady wind order reduction and c) developing a LSTM model to predict the blade deformations from the wind time series reduced space. Although this work is interesting, and clearly, the authors put a lot of effort into it, it needs more work to be publishable. I commented on the article extensively and in detail, which you can find in the supplement. These are my general comments:*
* * *
**Q 2.1** *The structure of the paper can be improved. It doesn't necessarily follow the chronological order. I think there needs to be a clearer separation between methodology and results. For example, Fig 4 can be in section 8.1.2, and the explanation about DCT needs to move the part where you explain DCT.*

**Reply**: We thank the reviewer for their valuable feedback on improving the structure of the paper. While we agree with the need for a clearer separation between methodology and results, we have chosen to retain Figure 4 in its current location to maintain the logical flow of the discussion. The figure is closely tied to

[Figure]

Figure 1: Rotor speed predictions at higher turbulence intensity

[Figure]

Figure 2: Blade pitch angle predictions at higher turbulence intensity

the methodological explanation of the 2D Discrete Cosine Transform (DCT), and relocating it to the results section could disrupt the reader's understanding of the process. However, we have revised the manuscript to consolidate the DCT explanation within the methodology section and removed redundant information to enhance clarity and readability. We believe these changes address the reviewer's concerns while preserving the coherence of the paper.
* * *
*Q 2.2 The article is well-written, but it could be more concise. Some sentences are long and written in passive voice, making them hard to follow.*

**Reply**: We thank the reviewer for their constructive feedback on improving the readability of the manuscript. In response, we thoroughly reviewed the text and revised it to reduce sentence length and convert passive voice to active voice whenever possible. We have also removed redundant information not relevant to this study to make the article more concise. We believe that these edits improve the overall readability of the manuscript.
* * *
*Q 2.3 The quality of the figures can be improved. Figures with time series data are small and not legible. The flowcharts can have a better graphical quality.*

**Reply**: We thank the reviewer for their feedback regarding the quality of the figures. We have revised the manuscript to improve the resolution and legibility of the time series figures wherever possible. However, for some figures, the quality is constrained by the nature of the data and the original source, making further enhancements challenging.
* * *
*Q 2.4 It is necessary to show how the LSTM results match OpenFAST and how much data is lost when reconstructing your wind time series from the reduced space (PCA or DCT).*

**Reply**: We thank the reviewer for their comment. The primary aim of this study is not to reconstruct wind time series using PCA or DCT but to use these methods as feature extraction techniques for the LSTM algorithm. The selection of principal components and discrete cosine transforms is guided by their relevance to the quantity of interest and the requirements of the machine learning algorithm, rather than reconstruction accuracy. As such, the focus is on leveraging these transformations to enhance the LSTM's predictive performance, not on minimizing reconstruction loss. Therefore, analysing the reconstruction error is beyond the scope of this study. Also, since the LSTM model is trained using the data from numerical model as the ground truth, direct comparison of LSTM model with OpenFast is not presented in the article. A more detailed response to this query is also included in the attached document.
* * *
*Q 2.5 The DCT part needs to be explained better. What do you do with your time when you are going through DCT? Section 8.1.2 makes this a bit clearer, but it needs to be explained earlier.*

**Reply**: We thank the reviewer for their comment regarding the explanation of the DCT method. At each time step, the wind field, modeled as a 25x25 grid, is decomposed into a selected number of DCT components. This process captures both spatial and temporal variations in the wind field, as the DCT is applied sequentially across time steps. The resulting components serve as feature inputs to the LSTM model, enabling it to learn the underlying dynamics of the wind turbine blade response. To address the reviewer's concern, we have enhanced the explanation of the DCT process earlier in the manuscript, ensuring a clearer and more comprehensive description of its role in the methodology. Additional details are also provided in the attached document for further clarification.

---

## Referee Report (RR1)

**Reviewer Report**

**Multi-task Learning Long Short-term Memory Model to Emulate Wind Turbine Blade Dynamics (Shubham Baisthakur and Breiffni Fitzgerald).**

The article presents a method of developing LSTM models for predicting blade response using multi-stage modelling and multi-task learning with dimensionality reduction techniques. This is a very well written article which presents a novel and relevant contribution in data driven methods for modelling blade dynamics. The authors have provided a satisfactory response to previous reviewer comments. I recommend the article to be published as it is, and have no further comments.

---

## Referee Report (RR2)

[referee-annotated manuscript omitted]

---

## Author Response (AR2)

**Response to Reviewer Comments**

**Shubham Baisthakur and Breiffni Fitzgerald**

**June 10, 2025**

We sincerely thank the reviewer for their thorough review of our manuscript titled *"Multi-task Learning Long Short-term Memory Model to Emulate Wind Turbine Blade Dynamics"* and for providing valuable feedback. The reviewer's comments have greatly contributed to improving the overall quality of this paper.

This document presents our responses to the reviewers' comments. Each comment is shown in *italic text*, followed by our response in regular text. Since Reviewer 2 provided comments directly on the manuscript rather than in a formal report, we have listed those queries here with corresponding page and line numbers to facilitate easier tracking. Additionally, a separate PDF file is attached to address Reviewer 2's detailed comments on the manuscript

**Reviewer 1**

*Q 1.1 The article presents a method of developing LSTM models for predicting blade response using multistage modelling and multi-task learning with dimensionality reduction techniques. This is a very well written article which presents a novel and relevant contribution in data driven methods for modelling blade dynamics. The authors have provided a satisfactory response to previous reviewer comments. I recommend the article to be published as it is, and have no further comments.*

**Reply**: We thank the reviewer for their positive evaluation and recommendation. We are pleased that the novelty and clarity of the proposed framework were appreciated. The reviewers' comments throughout the process have greatly helped improve the quality of the manuscript and are sincerely acknowledged.

**Reviewer 2**

*Q 2.1 Page 2, line 26: What about the accuracy?*

**Reply**: We appreciate the reviewer's attention to accuracy, which is indeed critical in any surrogate modelling approach. While the introduction is intended to broadly set the context and highlight the computational motivation, we have thoroughly addressed accuracy throughout the paper. Specifically, the model's accuracy is benchmarked against a validated numerical model in Section 7, with results showing consistent performance across all key response variables. To avoid ambiguity, we have now added a clarifying sentence at the end of the introductory paragraph.

*Q 2.2 Page 3, line 66: If you are using a sequential ML. If the model has a systematic approach, this statement is not valid.*

**Reply**: The statement in question is intended as a practical observation about the computational cost of training separate surrogate models for each DOF. Immediately following this statement, the manuscript proposes multi-task learning as an efficient solution that overcomes this challenge by predicting multiple outputs within a single model. Therefore, within the full context, the original statement is valid and serves to motivate our approach. We believe this logical flow is clear in the current manuscript.

***Q 2.3*** *Page 3, line 88: So, is it on monoplie? Jacket? floater? not clear*

**Reply**: Thank you for the comment. To clarify, the numerical model of the IEA-15MW wind turbine in this study is based on the monopile foundation. The manuscript is updated to explicitly state the foundation type for clarity.
* * *
***Q 2.4*** *Page 4, line 93: How many nodes do you have on the blade?*

**Reply**: Thank you for pointing this out. Although the blades are modelled using the modal summation method, which is not directly a node-based method, the aerodynamic and inertial forces are computed at 50 discrete nodes along the blade span, consistent with the original discretisation used by NREL in the IEA-15MW reference turbine definition. This detail is reflected in the updated manuscript.
* * *
***Q 2.5*** *Page 5, line 127: a) At which location along the blade? b) can you elaborate on the phase change for the in-plane displacement at 7m/s?*

**Reply**: a) Figures 1 and 2 have been updated to explicitly indicate that they present the displacement response at the blade tip. b) We thank the reviewer for pointing this out. A slight phase offset is indeed observed at 7m/s between the in-plane displacement responses of the numerical model and OpenFast. This deviation is localised to this specific operating condition and remains within an acceptable range across the overall response duration. Such minor phase differences may arise due to subtle differences in the numerical implementation of the structural damping model, integration schemes, or aerodynamic force evaluations between the two solvers. While our model and OpenFast follow the same underlying physical formulations, differences in solver architecture can introduce small discrepancies in time-domain comparisons. Importantly, this phase shift does not significantly affect the amplitude or frequency content of the response, nor the fidelity of the dataset used for training the surrogate model.
* * *
***Q 2.6*** *Page 8, Figure 4: Why not fore-aft? Side to side is directly coming from the rotor torque, while fore-aft is coming from the thrust. I think both are necessary. Also TSS is not defined before.*

**Reply**: While both directions are dynamically relevant, the side-to-side tower response is typically used for FFT-based validation in literature, as it is lightly damped and thus exhibits clearer spectral peaks corresponding to the structural modes. In contrast, the fore-aft response being more heavily damped due to aerodynamic damping tends to exhibit a flatter FFT spectrum, making it less informative for identifying natural frequencies. For this reason, we presented the side-to-side FFT to verify that the natural frequencies of the numerical model are captured accurately. Also, the TSS terminology is removed from the figure.
* * *
***Q 2.7*** *Page 9, Section 3.2: Your answer to my comment: This section needs clarification. What is happening to time here? TurbSim output is time series, so X(i,j) are time series! It is not clear what are you trying to do here is very helpful to understand this work better. I suggest to add it to the text.*

**Reply**: Thanks for your suggestion. The manuscript is updated to reflect these details.
* * *
***Q 2.8*** *Page 10, line 212: Be consistent. In figure 7 you are using y and z. Here, which one is vertical, and which one is horizontal. I assume y is vertical and x is horizontal. If that is true, why you have higher wind speed at the right hand side of figure 5. If that is not true, I think you need to rotate figure 5. I commented this to the previous review. Also, if x (or z) is the vertical axis, the reason for higher wind speed is shear too, not only turbulence.*

**Reply**: Thank you for highlighting this point again. We have revised Figure 7 to ensure consistency with the axis nomenclature used throughout the manuscript. As correctly noted, in our convention, the x-axis represents the horizontal direction and the y-axis denotes the vertical direction.

Regarding the higher wind speeds observed on the right-hand side of Figure 5, we agree that both shear and turbulence contribute to this pattern. While vertical wind shear typically manifests in a time-averaged sense the snapshot shown in Figure 5 captures the wind field at a single time instance, where turbulent

fluctuations can momentarily dominate and distort the expected shear profile. Therefore, the observed spatial variation in wind speed reflects a combined effect of transient turbulence and wind shear.
* * *
**Q 2.9** *Page 11, line 233: You answer my comment about high frequency component before. However, those high frequencies cause fatigue and by removing them your fatigue prediction is not valid anymore. I think this need to be mentioned.*

**Reply**: Thank you for your comment. To clarify, the high-frequency components are removed only from the wind field representation, not from the structural response. The DCT-based compression is used as a dimensionality reduction step for the input (wind field), with the goal of retaining dominant spatial patterns that generalise well across varying wind conditions. The full-resolution response data, which inherently contains higher-frequency content due to turbulence-induced effects, is retained and used during training. As evidenced in the results, the LSTM model is able to capture the dynamics in the blade displacement response, suggesting that the retained wind features are still informative for fatigue-relevant behaviour. The author intend to further investigate the impact of such models on fatigue analysis in a separate study.
* * *
**Q 2.10** *Page 15, line 332: How did you decide on these bounds? If you Why the lower bound for shear is zero?*

**Reply**: The wind speed range of 3–25 m/s is adopted from the operational range of the reference 15 MW wind turbine, ensuring that all simulations remain within its cut-in and cut-out wind speed limits. The bounds for turbulence intensity (TI) and the shear exponent are based on the approach outlined in:

- Dimitrov, N., Kelly, M. C., Vignaroli, A., & Berg, J. (2018). From wind to loads: wind turbine site-specific load estimation with surrogate models trained on high-fidelity load databases. Wind Energy Science, 3(2), 767-790.

Regarding the lower bound for the shear exponent, we chose 0 as the minimum rather than allowing negative shear values. The negative shear were excluded in this study to reduce complexity and limit the parameter space. Negative shear introduces atypical inflow conditions that can lead to unusual blade loading patterns, which, while interesting, were outside the scope of the present surrogate modelling framework focused on representative and commonly occurring operating scenarios.
This explanation is added to the manuscript for further clarity.
* * *
**Q 2.11** *Page 16, line 349: How did you decide on your mean wind speed? Why 16? How many simulations did you end up with?*

**Reply**: Thank you for your comment. As detailed in the manuscript, the range of mean wind speeds was chosen based on the operational wind speed range provided in the turbine properties. Within this range, 50 mean wind speed values were sampled using a Sobol sequence method. The selection of 16 seeds per mean wind speed point each representing a unique combination of turbulence intensity, wind shear exponent, and random seed was made by considering IEC recommendations alongside data augmentation needs, as explained in Section 5.1. Consequently, the dataset comprises 50 (wind speeds)×16 (realisations)=800 simulations.
We have updated the manuscript to clarify these details for the reader's benefit.
* * *
**Q 2.12** *Page 19,line 438: This is an interesting finding, and worth discussion. Why this is happening?*

**Reply**: Thank you for your comment. However, this observation was discussed in detail during the previous round of reviews. For clarity, we briefly restate the rationale here:
PCA and DCT both reduce dimensionality of the wind field but differ in how they capture patterns. The blade pitch angle is highly sensitive to localised spatial variations, such as shear and turbulence, which are better captured by DCT's spatial frequency decomposition. In contrast, rotor speed reflects a temporally integrated effect of the inflow, which aligns better with the global variance captured by PCA. This explains the observed performance distinction. We believe this explanation is sufficient for the scope of this study, but acknowledge that a more granular comparison may be pursued in future work.
The manuscript is updated for further clarity.

**Q 2.13** *Page 20,line 464: I commented this before. What are the hyperparameters and how did you end up with these bounds? You answered my comment, but I don't see it reflected in the text.*

**Reply**: We have now updated the manuscript to include this explanation.
* * *
**Q 2.14** *Page 20, line 468: What is rotor average wind speed?*

**Reply**: Thank you for your comment. Rotor averaged wind speed refers to the mean wind speed over the entire rotor swept area at a given time instant. This term is commonly used in wind energy research, which is why it was not explicitly defined in the manuscript. However, to avoid any ambiguity, we have included a brief definition in the revised manuscript for clarity.
* * *
**Q 2.15** *Page 21, Figure 10: I had two comments on this part before, which are not answered. The first one about training/testing/validation seperation, you answered, but it is not in the text. Which I think is necessary. The second one about RPM above rated and the fact that it is constant, I don't see any discussion or comment on that. I copy paste my comment here: So, this is not surprising. The results you are showing here is the rotor speed. This is mainly depends on average wind speed, which goes in directly into the model. The fluctuation in the wind specially the higher PCA mode wouldn't effect the rotor speed that much. Also, the results that you are showing here they are all above rated, which means the rotor speed is constant. I think it is necessary to show below rated results too.*

**Reply**: Thank you for highlighting the missing information regarding the data splitting approach, this has now been added to the updated manuscript.

The authors acknowledge the reviewer's observation, and the fact that the results are not surprising further highlights the interpretability of the model and the underlying physical behaviour of the system. These results also demonstrate the ability of the chosen feature selection method to uncover relevant features.

With regard to the comment on presenting results for below-rated wind conditions, we would like to reiterate the explanation already provided in Lines 448–451 of the manuscript for clarity: As previously mentioned, wind turbine dynamics differ significantly between above-rated and below-rated operational regimes. Consequently, models developed for each regime may require distinct architectures and input features to accurately capture the underlying physics. Presenting a second set of results for below-rated conditions including separate feature selection and hyperparameter tuning was deemed to introduce excessive detail without proportionate benefit to the core methodology. Nevertheless, the presented modelling framework remains applicable to both above- and below-rated regions.
* * *
**Q 2.16** *Page 21, Equation 21: This Z is DCT Z, right?*

**Reply**: Thank you for pointing this out. This typo is corrected in the updated manuscript.
* * *
**Q 2.17** *Page 23, Figure 13: What is on y axis?*

**Reply**: The y-axis represents the magnitude of DCT component mentioned in the subfigure title. This figure is updated in the revised manuscript.
* * *
**Q 2.18** *Page 26, line 535: What do you mean by constant feature here? They are not changing with time?*

**Reply**: Thank you for your comment. By "constant features," we refer to features that remain permanently included in the input feature set during the recursive feature addition process. This clarification has now been added to the manuscript.
* * *
**Q 2.19** *Page 26, line 545: Here you talk about DOF, which is necessary. However, when you present the data in terms of displacement in E and F directions, reader needs to know at which location on the blade.*

**Reply**: Thank you for this observation. To clarify the relationship between the DOF and the corresponding deformation response, we have highlighted the relevant nomenclature in Section 6. Additionally, Equation (20) defines the mapping between the DOFs and the displacement response.

***Q 2.20*** *Page 28, Figure 18: You space is limited and your plots are small. Instead of showing 600sec of results, which not that much can be observed, show 100sec. I can not really see what is happening in the edgewise direction.*

**Reply**: We thank the reviewer for the suggestion. Presenting the response over a 600-second duration follows a standard convention in wind turbine dynamics studies, as it demonstrates the model's robustness and ability to capture variations across the full turbulent inflow period. While a shorter time window may offer improved visual clarity, it risks overlooking long-term trends or transient behaviours that can emerge only over extended periods. That said, we acknowledge the need for better visual interpretation and have now split the original figure into two separate plots (Figures 18 and 19, Figures 20 and 21) for flapwise and edgewise responses. This revision ensures that the edgewise behaviour, which was previously harder to interpret, is now more clearly visible at the same time the validity of model results for the entire simulation duration is demonstrated.
* * *
***Q 2.21*** *Page 29, line 560: I asked this question before, I didn't get an answer. How your numerical model performs agains openfast model in unsteady wind? I think this question rises, as you try to answer to solve two problems:*

    *a Try to build a cheap model that replaces expensive openfast.*

    *b Try to build a surrogate model on a set of data using ML methods.*

*However, I think you aim for this paper is showing how this LSTM architecture can work. If that is correct, I think you need to emphasis on this more at the motivation part, and be more clear that the numerical method is nothing more than a dataset generator, and the reason you are using it is to bypass OpenFAST expensive simulations. Also, it is important to mention that the fact that this LSTM is trained and validated on a numerical model, and not OpenFAST is a limitation. The question that remains there, and can be answered in future work is how this works when you do this with OpenFAST model.*

**Reply**: We thank the reviewer for the comment. As discussed during the previous review stage, validation of the numerical model against OpenFast under steady wind conditions was deemed adequate for the objectives of this study. Furthermore, the numerical model is based on the same governing principles as OpenFast, including Kane's dynamics for deriving equations of motion, the modal summation method for modelling flexible elements, blade element momentum theory for computing aerodynamic loads, and Morison's equation for hydrodynamic forces—ensuring strong physical fidelity. The accuracy and application of the numerical model across multiple domains have already been established in prior peer-reviewed studies, now cited in the manuscript:

1. Sarkar, S., & Fitzgerald, B. (2021). Use of Kane's method for multi-body dynamic modelling and control of spar-type floating offshore wind turbines. Energies, 14(20), 6635.

2. Sarkar, S., & Fitzgerald, B. (2022). Fluid inerter for optimal vibration control of floating offshore wind turbine towers. Engineering Structures, 266, 114558.

3. Sarkar, S., Chen, L., Fitzgerald, B., & Basu, B. (2020). Multi-resolution wavelet pitch controller for spar-type floating offshore wind turbines including wave-current interactions. Journal of Sound and Vibration, 470, 115170.

The focus of the manuscript is on the multi-stage modelling framework and the performance evaluation of the LSTM-based surrogate model through a multi-task learning approach, as emphasised throughout the manuscript and reflected in the title. This methodology remains valid irrespective of the source of the data, and the model can be directly applied to datasets generated using OpenFast without modifying the learning framework. Hence, we do not consider the use of the proposed numerical model as a limitation. The codes developed for the surrogate model are openly shared with the manuscript, and interested readers are welcome to retrain the model using data from OpenFast or other high-fidelity aeroelastic solvers, if desired.

Lastly, we believe the reviewer may have misunderstood the objective of this study. At no point do we suggest that the numerical model serves as a computationally cheaper alternative to OpenFast.

***Q 2.22*** *Page 29, Section 7.3: The picture that you paint here, without talking about training and RFA computational cost is not complete.*

**Reply**: The primary focus of this section is on the inference-time performance of the surrogate model, which is the relevant metric in applications requiring repeated simulations. The training and feature selection processes are one-time, offline operations that are not repeated for each simulation and thus do not contribute to the operational cost of the surrogate model. We believe this separation of training and inference cost is both appropriate and widely accepted in the field.

***Q 2.23*** *Page 31, Figure 20: There is a vlalue to extract mean and std from you testing, for both actual and prediction and show them on a scatter plot with a regression line. That will show how good is your fit from statistical point of view. If you think about it, we generate all of these displacement/loads time series in wind turbine engineering, but at the end we look at them from statistical point of view (mean, std, min, max), so it is important that those at least mean and std match as well.*

**Reply**: We thank the reviewer for the helpful suggestion. While we agree that statistical comparisons can offer additional insights, the focus of this study is on capturing the dynamic behaviour. Time-domain validation through direct comparison of time series responses and NRMSE metrics sufficiently demonstrates the model's accuracy for the current scope. These provide a direct evaluation of the surrogate model's ability to reproduce the temporal characteristics of the blade response. We acknowledge the value of comparing statistical summaries and will consider this in future work aimed at applications such as fatigue assessment.

***Q 2.24*** *Page 32, line 585: I saw deformation, not loads, please revise.*

**Reply**: Thanks for pointing this out. This is rectified in the updated manuscript.

***Q 2.25*** *Page 32, Section 8: Maybe "Summary and Conclusion"? Also, please add future work and limitation.*

**Reply**: The manuscript is updated to include these suggestions.